# CASTOR: CAUSAL TEMPORAL REGIME STRUCTURE LEARNING

## ABSTRACT

The task of uncovering causal relationships among multivariate time series data stands as an essential and challenging objective that cuts across a broad array of disciplines ranging from climate science to healthcare. Such data entails linear or non-linear relationships, and usually follow multiple a priori unknown regimes. Existing causal discovery methods can infer summary causal graphs from heterogeneous data with known regimes, but they fall short in comprehensively learning both regimes and the corresponding causal graph. In this paper, we introduce CASTOR, a novel framework designed to learn causal relationships in multivariate time series data composed of various regimes, each governed by a distinct causal graph. Through the maximization of a score function via the EM algorithm, CASTOR infers the number of regimes and learns linear or non-linear causal relationships in each regime. We demonstrate the robust convergence properties of CASTOR, specifically highlighting its proficiency in accurately identifying unique regimes. Empirical evidence, garnered from exhaustive synthetic experiments and two real-world benchmarks, confirm CASTOR's superior performance in causal discovery compared to baseline methods. By learning a full temporal causal graph for each regime, CASTOR establishes itself as a distinctly interpretable method for causal discovery from multivariate time series composed of various regimes.

## 1 INTRODUCTION

Multivariate Time Series (MTS) is a very common type of data in a wide variety of fields. Uncovering the causal relationships among MTS variables and understanding how they evolve over time is crucial in numerous fields, such as climate science and health care. Although randomized controlled trials are widely recognized as the definitive method for determining causal relationships (Hariton & Locascio, 2018; McCoy, 2017), they often present challenges in terms of cost, ethics, or feasibility. Consequently, a multitude of causal discovery approaches now focus on extracting causality from observational data sources (Löwe et al., 2022; Bussmann et al., 2021; Pamfil et al., 2020; Moraffah et al., 2021; Runge, 2018; Wu et al., 2020).

Yet, the task of temporal causal discovery can be challenging. Firstly, the causal relationships between variables in the real-world context can manifest in either linear or non-linear forms. Secondly, certain causal effects may be instantaneous, making the differentiation between cause and effect particularly challenging. Various research works, such as those introducing models like DYNOTEARS (Pamfil et al., 2020) and Rhino (Gong et al., 2022), have aimed to address these challenges. Nevertheless, a prevalent assumption in many existing methods is that time series observations originate from a single "domain/regime" governed by one causal graph. This assumption is often inadequate in real-world scenarios, such as Electroencephalographic (EEG) time series in epilepsy settings (Tang et al., 2021; Rahmani et al., 2023). In these cases, recordings exhibit several unknown regimes (non-seizure, pre-seizure, and seizure), each lasting at least a few seconds, and may reoccur multiple times during the recording. In such scenarios, the unknown regimes originate from different causal models on the same set of variables, each one is described by a different temporal causal graph. Previous works have approached the challenge of learning causal graphs from MTS with different regimes. In particular, CD-NOD, developed by Huang et al. (2020), tackles time series with various

regimes by detecting the variables with changing causal modules. While CD-NOD provides a summary graph capturing behavioral changes across regimes, it falls short in inferring individual causal graphs. The overall summary graph does not effectively highlight changes between regimes. It's crucial to emphasize that assessing the full temporal causal graph per regime yields more accurate results than relying on the summary graph. Another relevant work dealing with MTS composed of multiple regimes is RPCMCI (Saggioro et al., 2020). In this approach, Saggioro et al. (2020) learn a temporal graph for each regime. However, they focus initially on inferring only time-lagged relationships and require prior knowledge of the number of regimes and transitions between them. Overall, most of the existing methods make certain assumptions that may limit their applicability, such as a prior knowledge of the number of regimes, prior information about regime indices, which are often not readily available.

We introduce CASTOR (Figure 2), a framework designed to learn both linear and nonlinear temporal causal relationships from multivariate time series (MTS) composed of multiple regimes (each regime corresponds to MTS block), without necessitating prior knowledge of regime indexes or the total number of regimes involved. Using a predetermined window for index initialization (Figure 3), CASTOR maximizes a score function via EM algorithm to infer the number of regimes and learns linear or non-linear causal relationships inherent in each regime. To the best of our awareness, our framework represents an unprecedented initiative in the time series landscape for the simultaneous learning of causal graphs, regime numbers, and regime indices. Our key contributions include:

- In this paper, we introduce CASTOR, a score based method for temporal causal structure learning tailored for multivariate time series composed of various regimes.
- CASTOR employs the EM algorithm in conjunction with the NOTEARS (Zheng et al., 2020) penalization method to deduce regime partitions and infer causal graphs.
- Building upon the intuition of LIN (Liu & Kuang, 2023), we demonstrate that regime indices can indeed be recovered and the underlying causal structure is identifiable.
- We conduct extensive synthetic experiments with ablation studies to demonstrate the advantages of CASTOR and its robustness across different settings. Additionally, we demonstrate its performance compared to baselines in real-world discovery benchmarks.

## 2 FRAMEWORK

Consider the multivariate time series (MTS) $(X_t)_{t \in \mathcal{T}} = (X_t^i)_{i \in \mathbf{V}, t \in \mathcal{T}}$ where $|\mathbf{V}| = d$ number of components of MTS $(X_t)_{t \in \mathcal{T}}$ and $\mathcal{T}$ is time index set. The MTS $(X_t)_{t \in \mathcal{T}}$ is aligned with a temporal causal graph elucidating its generative process. Each component of the MTS at time t is represented by a singular node within the graph $\mathcal{G}$, leading us to define $\mathcal{G}$ as follows:

**Definition 1** *(Temporal Causal Graph, Assaad et al. (2022))* $\mathcal{G} = (V, E)$ *is the temporal causal graph associated with the MTS $(X_t)_{t \in \mathcal{T}}$ is a DAG and the set of vertices in that graph consists of the set of components $X^1, \ldots, X^d$ at each time $t \in \mathbb{N}$. The edges $E$ of the graph are defined as follows: variables $X_{t-\tau}^i$ and $X_t^j$ are connected by a lag-specific directed link $X_{t-\tau}^i \to X_t^j$ in $\mathcal{G}$ pointing forward in time if and only if $X^i$ at time $t - \tau$ causes $X^j$ at time $t$ with a time lag of $\tau > 0$ for $i = j$ and with a time lag of $\tau \geq 0$ for $i \neq j$.*

In the rest of our paper, we use the same notation introduced by Gong et al. (2022), where $\mathbf{Pa}_{\mathcal{G}}^i(< t)$ refers to the parents of node $i$ in $\mathcal{G}$ at previous time (lagged parents) and $\mathbf{Pa}_{\mathcal{G}}^i(t)$ to the parents at the current time $t$ (instantaneous parents). This notation emphasizes that CASTOR learns a complete temporal causal graph for each regime, allowing for differentiation between time-lagged and instantaneous links. To clarify our notation, if $X^i$ at time $t - \tau$ causes $X^j$ at time $t$ with a time lag of $\tau > 0$, we have an edge $X_{t-\tau}^i \to X_t^j$ and $X^i \in \mathbf{Pa}_{\mathcal{G}}^j(< t)$ and if $\tau = 0$, we have an edge $X_t^i \to X_t^j$ and $X^i \in \mathbf{Pa}_{\mathcal{G}}^j(t)$.

**Structural Equation Models (SEMs).** Schölkopf et al. (2021) For a MTS $(X_t)_{t \in \mathcal{T}}$, the associated SEM for a given temporal causal graph, denoted as $\mathcal{G}$ for additive noise models (ANMs):

$$X_t^i = g_i \left( \mathbf{Pa}_{\mathcal{G}}^i(< t), \mathbf{Pa}_{\mathcal{G}}^i(t) \right) + \epsilon_t^i, \tag{1}$$

where $g_i$ is a deterministic function and $\epsilon$ are mutually independent noise variables. SEM describes the causal relationships between MTS components given a causal graph $\mathcal{G}$. For the same MTS and

by assuming causal Markov property (Definition 4 in the Appendix C), we can further define a **causal graphical model (CGM)** by a distribution $p$ as follows:

$$p\left(\boldsymbol{X}_t \mid \mathcal{G}\right) = \prod_{i=1}^{d} p\left(X_t^i \mid \mathbf{Pa}_{\mathcal{G}}^i(<t), \mathbf{Pa}_{\mathcal{G}}^i(t)\right),\tag{2}$$

with $p\left(\boldsymbol{X}_t \mid \mathcal{G}\right)$, called observational distribution, capturing the joint probability between $X_t^i$ the components of $\boldsymbol{X}_t$, and $p\left(X_t^i \mid \mathbf{Pa}_{\mathcal{G}}^i(<t), \mathbf{Pa}_{\mathcal{G}}^i(t)\right)$, refering to the conditional distribution of $X_t^i$ given parents of $X_t^i$ in the temporal causal graph $\mathcal{G}$.
A MTS can exhibit a single regime (as assumed in prior works like Rhino (Gong et al., 2022) and DYNOTEARS (Pamfil et al., 2020)) or multiple regimes, as in our approach. Each regime can be seen as a MTS block with a minimum duration, denoted as $\zeta$. Regimes occur sequentially, with the condition that a second regime can only commence after a duration of at least $\zeta$ from the initiation of the preceding regime. This modeling is particularly relevant in contexts such as epilepsy, where EEG recordings may encompass distinct regimes (e.g., non-seizure, pre-seizure, and seizure), each lasting at least a few seconds.

**Definition 2** *(MTS with multiple regimes) We say that MTS $(\boldsymbol{X}_t)_{t\in\mathcal{T}} = (X_t^i)_{i\in\mathbf{V}, t\leq\mathcal{T}}$ is composed of $K$ disjoint regimes if it exists a unique time partition $\mathcal{E} = (\mathcal{E}_u)_{u\in\{1,...,K\}}$, such that: $\cap_{u=1}^{K}\mathcal{E}_u = \emptyset$, $\mathcal{T} = \cup_{u=1}^{K}\mathcal{E}_u$ and $\forall u \in \{1,...,K\}$, the MTS $(\boldsymbol{X}_t)_{t\in\mathcal{E}_u}$ is stationary and associated with a unique temporal causal graph $\mathcal{G}^u$.*

Notably, the regime $v$ (where $v = u + 1$) begins at least $\zeta$ samples after the start of regime $u$ and also persists for a minimum of $\zeta$ samples. Additionally, if regime $u$ reoccurs in the MTS, its duration in the second appearance is also no less than $\zeta$ samples. All the indices corresponding to the first and second appearances of this regime are stored in $\mathcal{E}_u$.
It is crucial to note that $\mathcal{G}^u$ are regime-dependent, meaning that graphs vary across different regimes i.e, $\mathcal{G}^u \neq \mathcal{G}^v$. In the rest of our paper, we note $\mathcal{G} = (\mathcal{G}^u)_{u\in\{1,...,K\}}$ the set of temporal causal graph associated with a MTS $(\boldsymbol{X}_t)_{t\in\mathcal{T}}$ composed of $K$ regimes. Within each regime $u$, the temporal causal graph $\mathcal{G}^u$, can be represented by a collection of adjacency matrices, collectively denoted as $\boldsymbol{G}_{\tau\in[|0:L|]}^u$ with a fixed maximum lag $L < \zeta$. In simpler terms, if $X_{t-\tau}^i \to X_t^j$, then the coefficient $[G_\tau^u]_{ij} \neq 0$; otherwise, it is 0. When $\tau = 0$, $G_0^u$ signifies instantaneous links within the temporal causal graph $\mathcal{G}^u$.
We now propose a novel functional form for SEM that incorporates linear or non-linear relations, instantaneous links and multiple regimes:

$$\forall u \in \{1,...,K\}, \forall t \in \mathcal{E}_u : X_t^i = g_i^u\left(\mathbf{Pa}_{\mathcal{G}^u}^i(<t), \mathbf{Pa}_{\mathcal{G}^u}^i(t)\right) + \epsilon_t^i,\tag{3}$$

where $g_i^u$ is a general differentiable linear or non-linear function and $\epsilon_t^i \sim \mathcal{N}(0,1)$, follows to a normal distribution. By assuming causal Markov property, we can define the associated CGM:

$$\forall u \in \{1,...,K\}, \forall t \in \mathcal{E}_u : p\left(\boldsymbol{X}_t \mid \mathcal{G}^u\right) = \prod_{i=1}^{d} p\left(X_t^i \mid \mathbf{Pa}_{\mathcal{G}^u}^i(<t), \mathbf{Pa}_{\mathcal{G}^u}^i(t)\right),\tag{4}$$

As mentioned earlier, in certain real-world scenarios, MTS consist of $K$ unknown regimes and cannot typically be represented by a single DAG. A new formulation of the distribution $p$ describing the CGM in such scenarios is as follows:

$$p\left(\boldsymbol{X}_t\right) = \sum_{u=1}^{K} \mathbb{1}_{\mathcal{E}_u}(t) \cdot p\left(\boldsymbol{X}_t \mid \mathcal{G}^u\right),\tag{5}$$

where $p\left(\boldsymbol{X}_t \mid \mathcal{G}^u\right)$ is specified in Equation (4), while $\mathbb{1}_{\mathcal{E}_u}(t)$ denotes the indicator function, defined as $\mathbb{1}_{\mathcal{E}_u}(t) = 1$ if $t \in \mathcal{E}_u$ and 0 otherwise.

Previous works assume prior knowledge of time partition $\mathcal{E}$ or report a summary causal graph (Huang et al., 2020), falling short of elucidating the full temporal causal graph. In the next section we will present CASTOR an optimization based method for temporal causal structure learning tailored for MTS with multiple regimes.

## 3 CASTOR: CAUSAL TEMPORAL REGIME STRUCTURE LEARNING

In this section, we present a novel approach designed to learn the set of temporal causal graphs $\mathcal{G} = (\mathcal{G}^u)_{u \in \{1,...,K\}}$ associated with MTS $(\boldsymbol{X}_t)_{t \in \mathcal{T}}$ composed of multiple regimes, the total number of regimes $K$, and the associated regime indexes $\mathcal{E} = (\mathcal{E}_u)_{u \in \{1,...,K\}}$. Sections 3.1 and 3.2 present solutions tailored to both linear and nonlinear contexts, respectively.

### 3.1 CASTOR FOR LINEAR CAUSAL RELATIONSHIPS

We first consider the case where the causal relationships between the components are linear. Based on Eq (3), the SEM can then be articulated as follows: $\forall u \in \{1,...,K\}, \forall t \in \mathcal{E}_u : \boldsymbol{X}_t = \boldsymbol{X}_t \boldsymbol{G}_0^u + \sum_{\tau=1}^{L} \boldsymbol{X}_{t-\tau} \boldsymbol{G}_\tau^u + \boldsymbol{\epsilon}_t$. From the causal Markov assumption (4) in the Appendix C and using Eq (5) we have :

$$\log p\left(\boldsymbol{X}_{0:|\mathcal{T}|}\right) = \sum_{t=0}^{|\mathcal{T}|} \log \sum_{u=1}^{K} \mathbb{1}_{\mathcal{E}_u}(t) \cdot p\left(\boldsymbol{X}_t \mid \mathcal{G}^u\right), \tag{6}$$

Our objective is to learn simultaneously $K$, $\mathcal{E} = (\mathcal{E}_u)_{u \in \{1,...,K\}}$ and $\mathcal{G}^u = \boldsymbol{G}_{\tau \in [|0:L|]}^u, \forall u \in \{1,...,K\}$ that maximise the $\log p\left(\boldsymbol{X}_{0:|\mathcal{T}|}\right)$ defined in Eq (6). Estimating the aforementioned parameters $(K, \mathcal{E}$ and $\mathcal{G}^u, \forall u \in \{1,...,K\})$ is challenging as the regime indices remain unknown. The presence of the sum within the log terms in Eq (6) renders the log likelihood intractable. The Expectation-Maximization (EM) algorithm (Dempster et al., 1977) is well-suited to address such problems; it introduces new variables $\gamma_{t,u}$ to model regime participation, thereby transforming the log likelihood to the following form: $\log p\left(\boldsymbol{X}_{0:|\mathcal{T}|}\right) = \sum_{t=0}^{|\mathcal{T}|} \sum_{u=1}^{K} \gamma_{t,u} \log \left(\mathbb{1}_{\mathcal{E}_u}(t) \cdot p\left(\boldsymbol{X}_t \mid \mathcal{G}^u\right)\right)$. The term within the logarithm may equal zero, leading to divergence in our log-likelihood. To address this issue, we employ the soft-max function $\pi_{t,u}(\alpha) = \frac{\exp(\boldsymbol{\alpha}_{u,1} t + \boldsymbol{\alpha}_{u,0})}{\sum_{u=1}^{K} \exp(\boldsymbol{\alpha}_{u,1} t + \boldsymbol{\alpha}_{u,0})}$ as introduced by Samé et al. (2011). The log-likelihood is then expressed as follows: $\log p\left(\boldsymbol{X}_{0:|\mathcal{T}|}\right) = \sum_{t=0}^{|\mathcal{T}|} \sum_{u=1}^{K} \gamma_{t,u} \log \left(\pi_{t,u}(\alpha) \cdot p\left(\boldsymbol{X}_t \mid \mathcal{G}^u\right)\right)$. Also learning the temporal causal graphs $\mathcal{G} = (\mathcal{G}^u)_{u \in \{1,...,K\}}$, concurrently entails the estimation of the regime distribution $p\left(\boldsymbol{X}_t \mid \mathcal{G}^u\right)$. In line with the definition of CGM in Equation (2), we model CASTOR's joint density of the uth regime by:

$$f^u\left(\boldsymbol{X}_t\right) := \prod_{i=1}^{d} f_i^u\left(\mathbf{Pa}_{\mathcal{G}^u}^i(<t), \mathbf{Pa}_{\mathcal{G}^u}^i(t)\right), \tag{7}$$

where $f^u$ is a distribution family. It is important to highlight that while $f^u$ can, in theory, be any distribution, in this particular study, we focus on Gaussian noise, used by many works and for which they showed the identifiability of causal graphs for one regime (Huang et al., 2020; Peters & Bühlmann, 2014). As a result, our distribution $f^u$ will be a normal distribution.

#### 3.1.1 MAXIMIZATION STEP: GRAPH LEARNING

To learn the regime indexes $\mathcal{E} = (\mathcal{E}_u)_{u \in \{1,...,K\}}$ and the number of regimes $K$, CASTOR initially divides the MTS into $N_w > K$ equal time windows in the initialisation step, where each window represents one regime. CASTOR uses this initialisation and the formulation (7) to estimate the temporal causal graphs by maximising the following equation:

$$\sup_{\theta, \alpha} \frac{1}{|\mathcal{T}|} \sum_{u=1}^{N_w} \sum_{t=0}^{|\mathcal{T}|} \gamma_{t,u} \log \pi_{t,u}(\alpha) f^u\left(\boldsymbol{X}_t\right) - \lambda |\mathcal{G}^u|, \text{ s.t., } \boldsymbol{G}_0^u \text{ is a DAG},$$

$$\iff \begin{cases} \max_\alpha \frac{1}{|\mathcal{T}|} \sum_{u=1}^{N_w} \sum_{t=1}^{|\mathcal{T}|} \gamma_{t,u} \log\left(\pi_{t,u}(\alpha)\right) \\ \sup_\theta \frac{1}{|\mathcal{T}|} \sum_{u=1}^{N_w} \sum_{t \in \mathcal{E}_u} \log f^u\left(\boldsymbol{X}_t\right) - \lambda |\mathcal{G}^u|, \text{ s.t., } \boldsymbol{G}_0^u \text{ is a DAG}, \end{cases} \tag{8}$$

where $\theta$ stands for $\forall u \in \{1,...,K\} : \boldsymbol{G}_{[|0:L|]}^u$, $|\mathcal{G}^u|$ is the number of edges in the temporal causal graph of regime $u$. The first term is the averaged log-likelihood over data, the second is a penalty term with positive small coefficient $\lambda$ that controls the sparsity constraint. We impose an acyclicity

constraint solely on the adjacency matrix $\boldsymbol{G}_0^u$ of instantaneous links, as the other adjacency matrices $\boldsymbol{G}_{[|1:L|]}^u$ are inherently acyclic by definition, because these matrices establish links between variables at time $t$ and their time-lagged parents at time $t - \tau$. From the Eq (8), We can note $\mathcal{S}$, the score function of CASTOR as follows:

$$\mathcal{S}(\mathcal{G}, \mathcal{E}) := \sup_\theta \frac{1}{|\mathcal{T}|} \sum_{u=1}^{N_w} \sum_{t \in \mathcal{E}_u} \log f^u(\boldsymbol{X}_t) - \lambda |\mathcal{G}^u|, \text{ s.t } \boldsymbol{G}_0^u \text{ is a DAG,} \tag{9}$$

To address the optimization challenge that incorporates the acyclicity constraints, we employ an augmented Lagrangian method Zheng et al. (2018); Pamfil et al. (2020); Brouillard et al. (2020); Liu & Kuang (2023). Our Eq (9) can be succinctly written as:

$$\min_\theta \frac{1}{|\mathcal{T}|} \sum_{u=1}^{N_w} \sum_{t=1}^{|\mathcal{T}|} \gamma_{t,u} \left\| \boldsymbol{X}_t - \left( \boldsymbol{X}_t \boldsymbol{G}_0^u + \sum_{\tau=1}^{L} \boldsymbol{X}_{t-\tau} \boldsymbol{G}_\tau^u \right) \right\|_F^2 + \lambda |\mathcal{G}^u| + \frac{\rho}{2} h\left(\boldsymbol{G}_0^u\right)^2 + \alpha h\left(\boldsymbol{G}_0^u\right), \tag{10}$$

where $\alpha, \rho$ characterize the strength of the DAG penalty. The function $h(\boldsymbol{G}) = \text{tr}\left(e^{\boldsymbol{G} \odot \boldsymbol{G}}\right) - d$ corresponds to the acyclicity constraints proposed in Zheng et al. (2018) ($\odot$ is the Hadamard product).

### 3.1.2 EXPECTATION STEP: REGIME LEARNING

After this initial step of learning the graphs with $N_w$ equal windows, our method alternates between updating regime indexes $\mathcal{E} = (\mathcal{E}_u)_{u \in \{1, \dots, N_w\}}$, during the expectation phase (E-step), and inferring the temporal causal graphs $\boldsymbol{G}_{[|0:L|]}^u$ during the maximisation phase (M-step). In the expectation step, CASTOR updates the probability $\gamma_{t,u}$, that refers to the probability of $\boldsymbol{X}_t$ belonging to regime $u$, using the following equation (derivation details in Appendix B):

$$\gamma_{t,u} = \frac{\pi_{t,u}(\alpha) f^u(\boldsymbol{X}_t)}{\sum_{j=1}^{N_w} \pi_{t,j}(\alpha) f^j(\boldsymbol{X}_t)} \tag{11}$$

In the E-step, for each time sample $t$, CASTOR assigns a value of 1 to the most probable regime $u$ (with the highest $\gamma_{t,u}$), and 0 to others. Additionally, CASTOR filters out regimes with insufficient samples (fewer than $\zeta$, the minimum regime duration, defined as a hyper-parameter). Discarded regime samples are then reassigned to the nearest regime in terms of probability in the subsequent iteration (if $\boldsymbol{X}_t$ belongs to the discarded regime $u$, then it will be allocated to regime $v$ with the highest $\gamma_{t,v}$). Figure 3 illustrates the regime learning process.

### 3.2 CASTOR FOR NON-LINEAR CAUSAL RELATION

We now describe our approach to discerning non-linear causal relationships from MTS, starting with the case $K = 1$ for ease of description. Subsequently, we will outline how this methodology extends to address MTS with multiple regimes. In this setting, the SEM is the same as the one expressed in Eq (1): $X_t^i = g_i\left(\mathbf{Pa}_\mathcal{G}^i(<t), \mathbf{Pa}_\mathcal{G}^i(t)\right) + \epsilon_t^i$, where $g_i$ represents a differentiable function that carries the non linearity property for the causal relationships and $\epsilon_t^i \sim \mathcal{N}(0, 1)$. The associated CGM is defined similarly to Eq (2).

Our objective is to recover the temporal causal graph $\mathcal{G} = \boldsymbol{G}_{[|0:L|]}$, which is equivalent to estimate the true distribution defined in Eq (2): $p(\boldsymbol{X}_t \mid \mathcal{G}) = \prod_{i=1}^d p\left(X_t^i \mid \mathbf{Pa}_\mathcal{G}^i(<t), \mathbf{Pa}_\mathcal{G}^i(t)\right)$. We employ Neural Networks (NN) to accommodate the non-linearity introduced in our problem formulation, similarly to what is done in Brouillard et al. (2020); Liu & Kuang (2023), while utilizing NOTEARS (Zheng et al., 2018) constraints to enforce acyclicity for the graph modeling instantaneous links. The NN is used to estimate the parameters of our distribution, defined as[1]:

$$f(\boldsymbol{X}_t) := \prod_{i=1}^d f_i\left(\mathbf{Pa}_{\mathcal{G}^u}^i(<t), \mathbf{Pa}_{\mathcal{G}^u}^i(t)\right), \tag{12}$$

---

[1] Our work builds upon the foundational ideas presented in Zheng's work (Zheng et al. (2020)) on causal discovery in non-linear iid data.

with $f$ a normal distribution family. The parameters of the distribution, are determined by a neural networks NN based on variables' parents' value. For each time step $t$, we aggregate all time-lagged variables to form a single time lag vector, denoted as $\boldsymbol{X}_t^{\text{lag}} = [\boldsymbol{X}_{t-1}|\cdots|\boldsymbol{X}_{t-L}]$, which encompasses the lagged data. We employ $\boldsymbol{X}_t$ and $\boldsymbol{X}_t^{\text{lag}}$ as inputs for different neural networks $\text{NN}_i, \forall i \in \{1, ..., d\}$, with the objective of estimating the parameters of the distributions $f_i$. Mathematically, our Neural Networks are formulated as follows: $\forall i \in \{1, ..., d\} : \text{NN}_i(\boldsymbol{X}_t, \boldsymbol{X}_t^{\text{lag}}) = \psi_i\left(\phi_i(\boldsymbol{X}_t), \phi_i^{\text{lag}}(\boldsymbol{X}_t^{\text{lag}})\right)$, where $\psi_i$ are neural networks composed of locally connected layers introduced by Zheng et al. (2020) and activation functions $\phi_i, \phi_i^{\text{lag}}$ $(i \in \{1, ..., d\})$ composed of linear layers and sigmoid activation functions. The locally connected layers help to encode the variable dependencies in the first layer. Let $\Theta_i$ and $\Theta_i^{\text{lag}}$ represent the parameters of the first layer for $\phi_i, \phi_i^{\text{lag}}$ respectively. For a given node $i$, the instantaneous and time-lagged interactions with another node $j$ can be succinctly captured by examining the norms of the corresponding columns $j$ in the weight matrices of the initial layers such that: $[\boldsymbol{G}_0]_{ij} = \|\Theta_i(\text{column j})\|_2, [\boldsymbol{G}_\tau]_{ij} = \left\|\Theta_i^{\text{lag}}(\text{column j})\right\|_2$.

As we know the matrix $\Theta_i^{\text{lag}} \in \mathbb{R}^{d \times dL}$, recovering lag matrices by the above equation requires a reshape formulation, we keep the notation above for simplicity. The objective is thus to learn $\Theta_i$ and $\Theta_i^{\text{lag}}$ that will encode the causal relationships by maximizing the following score function:

$$\mathcal{S}(\mathcal{G}) := \sup_\theta \frac{1}{|\mathcal{T}|} \sum_{t=1}^{|\mathcal{T}|} f(\boldsymbol{X}_t) - \lambda|\mathcal{G}|, \text{ s.t } \boldsymbol{G}_0 \text{ is a DAG}, \tag{13}$$

Given a loss function $\mathcal{L}$ such as least squares, maximising Eq (13) is equivalent to:

$$\min_{\theta, \mathcal{G}} \frac{1}{|\mathcal{T}|} \sum_{t=1}^{|\mathcal{T}|} \sum_{i=1}^{d} \mathcal{L}(X_t^i, \psi_i\left(\phi_i(\boldsymbol{X}_t), \phi_i^{\text{lag}}(\boldsymbol{X}_t^{\text{lag}})\right)) + \lambda|\mathcal{G}| + \frac{\rho}{2}h\left(\boldsymbol{G}_0\right)^2 + \alpha h\left(\boldsymbol{G}_0\right) \tag{14}$$

where $\theta$ are the neural networks parameters and $\mathcal{G}$ is a notation of $\boldsymbol{G}_{\tau \in [|0:L|]}$. For the acyclicity constraint, we add the augmented Lagrangian term $\frac{\rho}{2}h\left(\boldsymbol{G}_0\right)^2 + \alpha h\left(\boldsymbol{G}_0\right)$ to our optimization problem. Similarly to the linear case, CASTOR for non linear case uses the EM algorithm that alternates between distribution parameters learning (these parameters include the causal graphs) and regime learning. The sole change between the linear and non-linear scenarios resides in the methodology employed for estimating the distribution parameters. In the linear instance, we have demonstrated that the parameters are directly linked to the temporal causal graphs through the adjacency matrices. Conversely, in the non-linear scenario, we leverage Neural Networks (NNs) to estimate the distribution parameters. Hence, the Maximisation step of the EM algorithm, specifically tailored for the non-linear case, is as follows:

$$\min_{\theta, \mathcal{G}} \frac{1}{|\mathcal{T}|} \sum_{u=1}^{N_w} \sum_{t=1}^{|\mathcal{T}|} \sum_{i=1}^{d} \gamma_{t,u} \mathcal{L}(X_t^i, \psi_i^u\left(\phi_i^u(\boldsymbol{X}_t), \phi_i^{u,\text{lag}}(\boldsymbol{X}_t^{\text{lag}})\right)) + \lambda|\mathcal{G}^u| + \frac{\rho}{2}h\left(\boldsymbol{G}_0^u\right)^2 + \alpha h\left(\boldsymbol{G}_0^u\right)$$
$$\tag{15}$$

where $\theta$ summarises all the network parameters and $\mathcal{G} = (\mathcal{G}^u)_{u \in \{0, ..., K\}}$. The formulations of the E-step and regime adjustment remains the same as in the linear case. Algorithm (1) outlines our CASTOR model for both linear and nonlinear causal relationships. It details the process for updating parameters at each iteration and employs a minimum regime duration $\zeta$ to eliminate unnecessary regimes, thereby determining the optimal number of regimes.

---

**Algorithm 1** CASTOR algorithm

---

**procedure** CASTOR($\boldsymbol{X}, W, L, N_{\text{iter}}, \zeta$)                   ▷ $\boldsymbol{X} \in \mathbb{R}^{T \times d}$, $W$: window size
    **while** iter $\leq N_{\text{iter}}$ **do**
        $\gamma_{t,u} \leftarrow \frac{\pi_{t,u}(\alpha)f^u(\boldsymbol{X}_t)}{\sum_{j=1}^{N_w} \pi_{t,j}(\alpha)f^j(\boldsymbol{X}_t)}$                               ▷ E step 11
        $\alpha \leftarrow \arg\min_\alpha \sum_{u=1}^{K} \sum_{t=1}^{|\mathcal{T}|} \gamma_{t,u} \log\left(\pi_{t,u}(\alpha)\right)$           ▷ M step, 8
        $\mathcal{G} \leftarrow$ argmin of Eq 10 or 15                                 ▷ M step
        **if** $\sum_t^{|\mathcal{T}|} \gamma_{t,u} \leq \zeta$ **then**
            $\forall t : \gamma_{t,u} \leftarrow 0$                              ▷ Cancel the regime $u$
    **return** $\gamma, \mathcal{G}$

---

## 4 THEORETICAL GUARANTEES

In this section, we present the theoretical underpinnings that ensure the robustness of CAS-TOR as a causal discovery method for time series data encompassing multiple regimes. Specifically, we provide guarantees on two fronts: the unambiguous identification of regime indices, and the accurate inference of causal relationships. Together, these guarantees confirm the ability of CASTOR to uniquely discern causal relations across time series with varying regimes. CASTOR learns the temporal causal graphs by optimizing a score function (Eq (9): $\mathcal{S}(\mathcal{G}, \mathcal{E}) = \sup_\theta \frac{1}{|\mathcal{T}|} \sum_{u=1}^{N_u} \sum_{t \in \mathcal{E}_u} \log f^u(\boldsymbol{X}_t) - \lambda |\mathcal{G}^u|)$ during the M-step. However, the score function also depends on a time index partition $\mathcal{E}$, which CASTOR learns in the expectation step. Consequently, the convergence of CASTOR and its capability to accurately discern the true regime partition, as well as construct appropriate temporal causal graphs, are contingent upon effectively optimizing the score function until it attains its optimal value. Let $\mathcal{G}^*$ be the set of ground truth causal graphs i.e $\mathcal{G}^* = (\mathcal{G}^{*,u})_{u \in \{1,...,K\}} = (\boldsymbol{G}^{*,u}_{[|0:L|]})_u$ and $\mathcal{E}^*$ the correct partition based on regime index. The following theorem summarizes our main theoretical contribution.

**Theorem 1** *We assume that each regime has enough data and the penalty coefficients in Equation (15) are sufficiently small, and all the assumption 1, 2,3,4,5,6,7 hold (Appendix C for precise statements)), we have for any estimation $(\hat{\mathcal{G}}, \hat{\mathcal{E}})$ : $\mathcal{S}(\mathcal{G}^*, \mathcal{E}^*) > \mathcal{S}(\hat{\mathcal{G}}, \hat{\mathcal{E}})$, if $\exists u \in \{1,...,K\}$ s.t $\hat{\mathcal{G}}^u$ disagrees with $\mathcal{G}^{*,u}$ on instantaneous or/and time lagged link, or any regime $\hat{\mathcal{E}}_u \in \hat{\mathcal{E}}$ is close to none of the true regimes in the sense of Kullback–Leibler divergence.*

*Proof.* Details of the proof in the Appendix C.1.
The theorem indicates that inaccuracies in identifying causal structures or real regimes yield a suboptimal estimation score. By optimizing according to Equation (9) we will asymptotically identify the actual regimes and recover true causal graphs, leading to the subsequent corollary.

**Corollary 1.1** *Given the conditions stated in Theorem 1, if the score proposed in Equation (9) is optimized, then samples in each regime would approach one of the true regimes asymptotically in the sense of Kullback-Leibler divergence and the causal structure of each regime is identifiable.*

## 5 EXPERIMENTS

### 5.1 SYNTHETIC DATA

| $d$ | Method | Type | $K = 2$ SHD | F1 | $K = 3$ SHD | F1 | $K = 4$ SHD | F1 |
|---|---|---|---|---|---|---|---|---|
| 10 | VARLINGAM | Inst | 26 | $18.2_{\pm 3.8}$ | 39 | $11.0_{\pm 6.4}$ | 59 | $9.70_{\pm 1.8}$ |
| | | Lag | 16 | $10.4_{\pm 7.8}$ | 28 | $5.01_{\pm 1.4}$ | 33 | $5.10_{\pm 3.1}$ |
| | RPCMCI | Inst | - | - | - | - | - | - |
| | | Lag | 38 | $42.3_{\pm 11.1}$ | 41 | $18.8_{\pm 2.5}$ | - | - |
| | CASTOR | Inst | **0** | $\mathbf{100}_{\pm 0}$ | **1** | $\mathbf{97.3}_{\pm 2.5}$ | **1** | $\mathbf{99.3}_{\pm 0.9}$ |
| | | Lag | **0** | $\mathbf{100}_{\pm 0}$ | **0** | $\mathbf{100}_{\pm 0}$ | **2** | $\mathbf{98.0}_{\pm 1.4}$ |
| 40 | VARLINGAM | Inst. | 134 | $8.40_{\pm 1.2}$ | 202 | $9.83_{\pm 0.9}$ | 248 | $10.9_{\pm 3.1}$ |
| | | Lag | 102 | $1.2_{\pm 0.1}$ | 137 | $1.13_{\pm 0.8}$ | 225 | $1.43_{\pm 0.6}$ |
| | RPCMCI | Inst. | - | - | - | - | - | - |
| | | Lag | 155 | $42.1_{\pm 3.5}$ | 321 | $18.4_{\pm 14}$ | - | - |
| | CASTOR | Inst. | **2** | $\mathbf{98.3}_{\pm 1.7}$ | **9** | $\mathbf{98.2}_{\pm 1.2}$ | **7** | $\mathbf{98.3}_{\pm 0.4}$ |
| | | Lag | **0** | $\mathbf{100}_{\pm 0.0}$ | **1** | $\mathbf{99.8}_{\pm 0.2}$ | **2** | $\mathbf{98.9}_{\pm 0.9}$ |

Table 1: F1 and SHD Scores by Models and Settings: $d$ indicates number of nodes, $K$ refers to the number of regimes, and 'Type' refers to the causal links as either instantaneous or time-lagged.

We perform extensive experiments to evaluate the performance of our method, CASTOR, in both linear and non-linear causal relationships. For groundtruth graph generation, we employ the Barabási-Albert (Barabási & Albert, 1999) model with a degree of 4 to establish instantaneous links,

while we utilize the Erdős–Rényi (Newman, 2018) random graph model with degrees ranging from 1 to 2 for time-lagged relationships. We focus on scenarios with a single time lag $L = 1$, although additional experiments involving multiple lags are detailed in the Appendix. The duration of each regime is chosen randomly from the set $\{300, 400, 500, 600\}$. Each experiment (combination choice of $K$ and $d$) was repeated three times under multiple settings, all combinations yield to more than 60 different datasets.

**Linear Relationships.** We examine varying numbers of nodes, specifically $\{5, 10, 20, 40\}$, and generated time series with different regime counts $\{2, 3, 4, 5\}$. Our model's performance is benchmarked against multiple baselines, namely RPCMCI (Saggioro et al., 2020) and VARLINGAM (Hyvärinen et al., 2010) and the results are presented in Table 1. RPCMCI represents the sole baseline tailored to address a similar setting. RPCMCI necessitates specific parameters, including the number of regimes and the maximum number of transitions, and with this input, it only infers time-lagged relations. Even with this detailed information, RPCMCI struggles to achieve convergence, particularly in settings with more than 3 different regimes. In contrast, CASTOR does not only surpass RPCMCI in performance but also converges consistently, correctly identifying the number of regimes and recovering both the regime indices and the underlying causal graphs of each regime. We can notice that CASTOR successively infers the regime indexes and learns as well the instantaneous links (more than 95% F1 score in different settings) as well as time lagged relations (more than 97% in almost all the settings). When we compare CASTOR with VARLINGAM (which performs causal discovery method for MTS data which can model both lagged and instantaneous links), the former also demonstrates markedly superior performance. To provide context, we manually partition our generated data into discrete regimes to facilitate the evaluation of VARLINGAM. We then executed VARLINGAM on each segmented regime, synthesized the graphs, and compared these composite structures against their true counterparts. Even when executing VARLINGAM separately on each regime, CASTOR still surpasses VARLINGAM, all without access to any prior information, such as the number of regimes or the indices of the regimes. Additional results that confirm the above results are in the Appendix D.

**Non-linear Relationships.** For non-linear relationships, the functions $g_i^u$ defined in Eq (3) include random weights generated from a uniform distribution over the interval ]0,2], coupled with activation functions selected randomly from the set $\{\texttt{Tanh}, \texttt{LeakyReLU}, \texttt{ReLU}\}$. In this case, we compare our model against various baseline models, namely DYNOTEARS (Pamfil et al., 2020) and VARLINGAM (Hyvärinen et al., 2010) and conducted experiments with different numbers of regimes $\{2, 3, 4\}$ and nodes $\{10, 20\}$. We can see from Figure (1) that both CASTOR and DYNOTEARS exhibit superior performance to VARLINGAM. It is important to outline that DYNOTEARS and VARLINGAM are each applied to individual regimes separately; neither is designed to learn or infer the number or indices of regimes. CASTOR demonstrates comparable performance to DYNOTEARS in modeling time-lagged relations for non-linear scenarios while learning also the regime indexes. It also succeeds in identifying instantaneous links. DYNOTEARS achieves inferior results in identifying instantaneous links due to his formulation that takes into consideration only linear relationships. This is much clearer in high dimensions due to the increasing complexity of the problem. Additional experiments are available in Appendix D, and show the same trends as explained above.

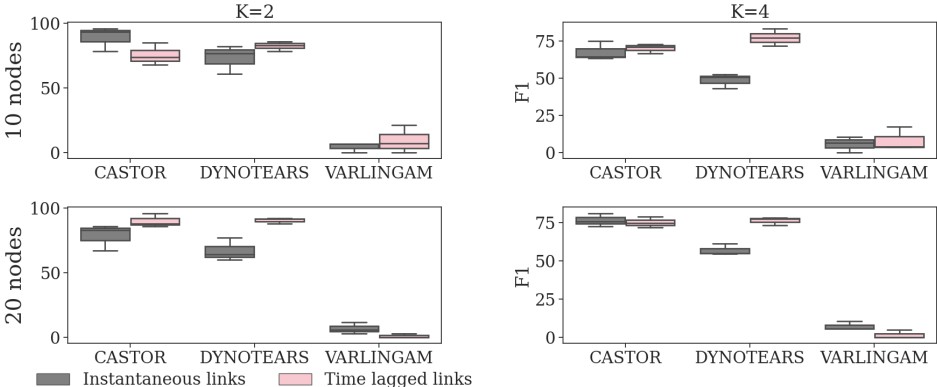

Figure 1: F1 by Models and Settings, Inst means instantaneous links and lag time-lagged ones.

## 5.2 WEB ACTIVITY DATASET

We now evaluate[2] CASTOR on two stacked IT monitoring time series, each comprising 1106 timestamps and 7 nodes, sourced from EasyVista [3]. These series capture activity metrics from a web server. We compared our method with PWGC (Granger Causality (Granger, 1969)) and VARLINGAM (Hyvärinen et al., 2010). As it is evident from Table 2, CASTOR proficiently identifies the exact number of regimes and regime indices. It is pertinent to note that PWGC and VARLINGAM are not inherently designed to

| Model | F1 Reg1 | F1 Reg 2 | Reg Acc |
|-----------|---------|----------|---------|
| PWGC | 53.8 | 20.0 | - |
| VARLINGAM | **66.7** | 0 | - |
| CASTOR | 18.2 | **28.5** | **100** |

Table 2: F1 Scores across Two IT Datasets: 'Reg' indicates regime and 'Reg Acc' refers to regime identification accuracy.

infer regime indices, we evaluate both models separately on each regime. On regime 2, CASTOR outperforms PWGC and VARLINGAM in learning causal relationships. While VARLINGAM exhibits superior results compared to CASTOR in regime 1, it is not designed to learn the causal graph and the indices. Also, given that our data pertains to IT monitoring, the likelihood of the presence of instantaneous links is relatively low, which could account for the good performance of VARLINGAM and PWGC that models only time-lagged links.

## 6 RELATED WORKS

**Causal structure learning from time series.** Assaad et al. (2022) offer an extensive survey of methods for learning temporal causal relationships. Most notably, Granger causality is the primary approach used for causal discovery from time series (Amornbunchornvej et al., 2019; Wu et al., 2020; Löwe et al., 2022; Bussmann et al., 2021; Xu et al., 2019). However, it is unable to accommodate instantaneous effects. DYNOTEARS (Pamfil et al., 2020), on the other hand, leverages the acyclicity constraint established by Zheng et al. (2018) to continuously relax the DAG and differentially learn instantaneous and time lagged structures. However, DYNOTEARS is still limited to linear functional forms. TiMINo (Peters et al., 2013) provides a general theoretical framework for temporal causal discovery with functional causal models and also a practical algorithm that learns casual relationships that can be non linear. However, the aforementioned methods assume that MTS are composed of a single regime.

**Causal structure learning from heterogeneous data.** Several studies have sought to tackle the challenge of causal discovery in heterogeneous data (Huang et al., 2020; Saeed et al., 2020; Zhou et al., 2022; Günther et al., 2023; Saggioro et al., 2020). Remarkably, Huang et al. (2020) address heterogeneous time series by modulating causal relationships through a regime index. While it provides a summary graph highlighting behavioral changes across regimes, they cannot infer individual causal graphs neither the exact number of regime. Meanwhile, LIN (Liu & Kuang, 2023) investigates the problem of causal structure learning from MTS with interventional data but in the absence of domain (observational or interventional) data. LIN cannot learn different graphs: it learns the indices of different domains and one causal graph (represents the instantaneous links) per MTS. Finally, Saggioro et al. (2020) assume knowledge of the number of regimes and propose the inference of only time-lagged links. Furthermore, they evaluate their algorithm on graphs with a limited number of nodes.

## 7 CONCLUSION

The task of inferring temporal causal graphs from observational time series is essential in numerous fields. It involves modeling linear and non-linear relationships and identifying multiple regimes, often without prior knowledge of regime indices. We introduce CASTOR, a new score-based with proven convergence properties, preventing the need for prior knowledge of regimes. Our method demonstrates superior performance in handling both linear and non-linear relationships across multiple regimes in synthetic and real datasets.

---

[2]Our evaluation methodology aligns with the approach outlined in Assaad et al. (2022)

[3]https://github.com/ckassaad/causal_discovery_for_time_series

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

## A ILLUSTRATIVE FIGURES OF CASTOR FRAMEWORK

CASTOR represents a causal discovery framework tailored for Multivariate Time Series (MTS), characterized by diverse regimes. In essence, CASTOR operates under the assumption that each MTS may be intricately composed of various unidentified regimes. Each regimes can be treated as an independent MTS. Additionally, it is crucial to note that the number of lags always remains below the minimum length of the regimes.

Figure (2) illustrates a MTS on its left side comprising three variables and two unknown distinct regimes. Each regime possesses its temporal DAG, with one lag attributed to each in this demonstrative scenario.

Upon receiving the MTS as input, CASTOR engages in the process of discerning the number of regimes, determining the indexes associated with each regime (indicating their commencement and conclusion), and inferring the temporal DAGs. The resultant DAGs facilitate the straightforward reconstruction of summary graphs encapsulating the entire MTS (CD-NOD output).

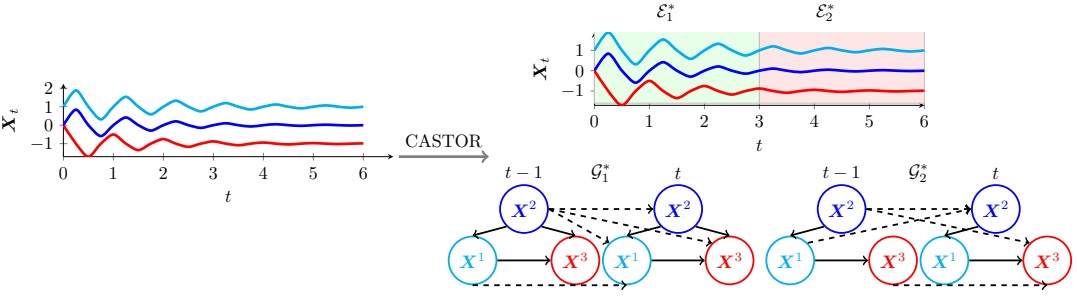

Figure 2: Overview of CASTOR. The dashed edges refer to the time lagged links, the normal arrows represent the instantaneous links.

To elucidate the intricacies of the regime learning process, Figure (3) delineates the step-by-step procedure followed by CASTOR in determining the number of regimes and their corresponding indexes. The process commences with CASTOR partitioning the Multivariate Time Series (MTS) into equal windows. In the initial iteration, the length of each regime equals the window size, a user-specified hyperparameter.

Subsequently, CASTOR embarks on learning a temporal DAG for each regime. This involves solving an optimization problem, as outlined in Eq(10) for the linear case and Eq(15) for the non-linear scenario. Following graph acquisition, CASTOR updates the regime indexes utilizing Eq(11). Notably, CASTOR employs a filtering mechanism to eliminate regimes characterized by an insufficient number of samples. In practical terms, any regime with fewer samples than a defined hyperparameter, denoted as $\zeta$ (representing the minimum regime duration), is discarded.

In the event of regime elimination, samples from the discarded regimes are reallocated to the nearest regime in terms of probability. Specifically, if the discarded regime is denoted as $u$, the sample $X_t$ will be assigned to regime $v$ in the subsequent iteration, where $v$ is the regime with the highest $\gamma_{t,v}$.

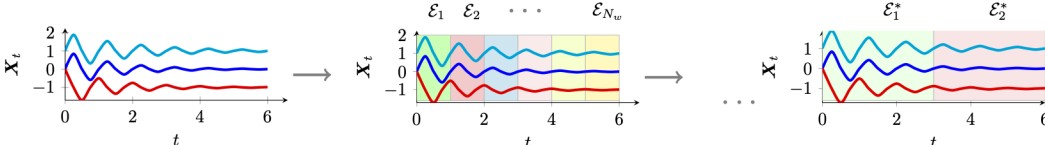

Figure 3: CASTOR's regime learning procedure

## B    EXPECTATION-MAXIMIZATION DERIVATION

In this section, we shall elucidate the computational details surrounding the resolution of our optimization problem. Specifically, we will provide clarity on the various equations introduced in Section 3, namely, Eq (8, 10, 11, 15).

**E-step.** We model regime participation through a binary latent variable $z_t \in \mathbf{R}^{N_w}$; $\boldsymbol{X}_t$ belongs to regime $u \Rightarrow z_{t,u} = 1$.

$$
\begin{aligned}
\gamma_{t,u} &= p\left(z_{t,u} = 1 \mid \boldsymbol{X}_t, \boldsymbol{G}^u_{\{0:L\}}\right) \\
&= \frac{p\left(z_{t,u} = 1\right) p\left(\boldsymbol{X}_t \mid z_{t,u} = 1, \boldsymbol{G}^u_{\{0:L\}}\right)}{\sum_{j=1}^{N_w} p\left(z_{t,j} = 1\right) p\left(\boldsymbol{X}_t \mid z_{t,j} = 1, \boldsymbol{G}^j_{\{0:L\}}\right)} \\
&= \frac{\pi_{t,u}(\alpha) f^u\left(\boldsymbol{X}_t\right)}{\sum_{j=1}^{N_w} \pi_{t,j}(\alpha) f^j\left(\boldsymbol{X}_t\right)}
\end{aligned}
\tag{16}
$$

**M-step.** Having estimated probabilities $\gamma_{t,u}$ in the E-step, we can now maximise the expected posterior distribution given the MTS $(\boldsymbol{X}_t)_{t \in \mathcal{T}}$ and we have:

$$
\begin{aligned}
&\sup_{\theta,\alpha} \frac{1}{|\mathcal{T}|} \sum_{u=1}^{N_w} \sum_{t=0}^{|\mathcal{T}|} \gamma_{t,u} \log \pi_{t,u}(\alpha) f^u\left(\boldsymbol{X}_t\right) - \lambda|\mathcal{G}^u|, \text{ s.t } \boldsymbol{G}^u_0 \text{ is a DAG}, \\
&\iff \begin{cases} \max_\alpha \frac{1}{|\mathcal{T}|} \sum_{u=1}^{N_w} \sum_{t=1}^{|\mathcal{T}|} \gamma_{t,u} \ln\left(\pi_{t,u}(\alpha)\right) \\ \sup_\theta \frac{1}{|\mathcal{T}|} \sum_{u=1}^{N_w} \sum_{t \in \mathcal{E}_u} \log f^u\left(\boldsymbol{X}_t\right) - \lambda|\mathcal{G}^u|, \text{ s.t } \boldsymbol{G}^u_0 \text{ is a DAG}, \end{cases}
\end{aligned}
\tag{17}
$$

We know $f^u\left(\boldsymbol{X}_t\right) = \mathcal{N}\left(\boldsymbol{X}_t \boldsymbol{G}^u_0 + \sum_{\tau=1}^{L} \boldsymbol{X}_{t-\tau} \boldsymbol{G}^u_\tau, I\right)$, hence:

$$
\iff \sup_\theta \frac{1}{|\mathcal{T}|} \sum_{u=1}^{N_w} \sum_{t \in \mathcal{E}_u} \log f^u\left(\boldsymbol{X}_t\right) - \lambda|\mathcal{G}^u|, \text{ s.t } \boldsymbol{G}^u_0 \text{ is a DAG},
$$

$$
\iff \min_\theta \frac{1}{|\mathcal{T}|} \sum_{u=1}^{N_w} \sum_{t \in \mathcal{E}_u} \left\| \boldsymbol{X}_t - \left( \boldsymbol{X}_t \boldsymbol{G}^u_0 + \sum_{\tau=1}^{L} \boldsymbol{X}_{t-\tau} \boldsymbol{G}^u_\tau \right) \right\|^2_F + \lambda|\mathcal{G}^u|, \text{ s.t } \boldsymbol{G}^u_0 \text{ is a DAG}
$$

$$
\iff \min_\theta \frac{1}{|\mathcal{T}|} \sum_{u=1}^{N_w} \sum_{t=1}^{|\mathcal{T}|} \gamma_{t,u} \left\| \boldsymbol{X}_t - \left( \boldsymbol{X}_t \boldsymbol{G}^u_0 + \sum_{\tau=1}^{L} \boldsymbol{X}_{t-\tau} \boldsymbol{G}^u_\tau \right) \right\|^2_F + \lambda|\mathcal{G}^u|, \text{ s.t } \boldsymbol{G}^u_0 \text{ is a DAG}
$$

$$
\iff \min_\theta \frac{1}{|\mathcal{T}|} \sum_{u=1}^{N_w} \sum_{t=1}^{|\mathcal{T}|} \gamma_{t,u} \left\| \boldsymbol{X}_t - \left( \boldsymbol{X}_t \boldsymbol{G}^u_0 + \sum_{\tau=1}^{L} \boldsymbol{X}_{t-\tau} \boldsymbol{G}^u_\tau \right) \right\|^2_F + \lambda|\mathcal{G}^u| + \frac{\rho}{2} h\left(\boldsymbol{G}^u_0\right)^2 + \alpha h\left(\boldsymbol{G}^u_0\right),
\tag{18}
$$

The only difference between the linear and the non linear cases is how we estimate the mean of the normal distribution $f^u$ for every regime $u$. As we mentioned in section 3.2, we estimate these means using NNs and we have $f^u_i\left(\boldsymbol{X}_t\right) = \mathcal{N}\left(\psi^u_i\left(\phi^u_i(\boldsymbol{X}_t), \phi^{u,\text{lag}}_i(\boldsymbol{X}^{\text{lag}}_t)\right), 1\right)$, Hence, our M-step for non-linear CASTOR:

$$
\min_{\theta,\mathcal{G}} \frac{1}{|\mathcal{T}|} \sum_{u=1}^{N_w} \sum_{t=1}^{|\mathcal{T}|} \sum_{i=1}^{d} \gamma_{t,u} \mathcal{L}(X^i_t, \psi^u_i\left(\phi^u_i(\boldsymbol{X}_t), \phi^{u,\text{lag}}_i(\boldsymbol{X}^{\text{lag}}_t)\right)) + \lambda|\mathcal{G}^u| + \frac{\rho}{2} h\left(\boldsymbol{G}^u_0\right)^2 + \alpha h\left(\boldsymbol{G}^u_0\right)
\tag{19}
$$

## C   REGIME AND CAUSAL GRAPHS IDENTIFIABILITY

In this section, we concentrate on establishing the identifiability of regimes and causal graphs within the CASTOR framework. Before diving into the details, let us set and clarify the required assumptions.

**Definition 3** *(Causal Stationarity Runge (2018)). The time series (that has one regime) process* $(\boldsymbol{X}_t)_{t \in \mathcal{T}}$ *with a graph* $\mathcal{G}$ *is called causally stationary over a time index set* $\mathcal{T}$ *if and only if for all links* $X_{t-\tau}^i \to X_t^j$ *in the graph*

$$X_{t-\tau}^i \not\perp\!\!\!\perp X_t^j \mid \boldsymbol{X}_{<t} \backslash \left\{ X_{t-\tau}^i \right\} \text{ holds for all } t \in \mathcal{T}$$

This elucidates the inherent characteristics of the time-series data generation mechanism, thereby validating the choice of the auto-regressive model. In our setting, we generalize Causal Stationarity as follows:

**Assumption 1** *(Causal Stationarity for time series with multiple regimes). The time series process* $(\boldsymbol{X}_t)_{t \in \mathcal{T}}$ *comprise multiple regimes* $K$, *where* $K$ *is the number of regime, we note* $\mathcal{E}_u = \{t | \gamma_{t,u} = 1\}$ *the set of time indexes where the regime* $u$ *is active, and* $\mathcal{T} = \cup_u \mathcal{E}_u$. $(\boldsymbol{X}_t)_{t \in \mathcal{T}}$ *with a graph* $\{\mathcal{G}^u\}_{u \in \{1,...,K\}}$ *is called causally stationary over a time index set* $\mathcal{T}$ *if and only if for all* $u \in \{1,...,K\}$, $(\boldsymbol{X}_t)_{t \in \mathcal{E}_u}$ *is causal stationary with graph* $\mathcal{G}^u$ *for time index set* $\mathcal{E}_u$.

**Definition 4** *(Causal Markov Property, Peters et al. (2017)). Given a DAG* $\mathcal{G}$ *and a joint distribution* $p$, *this distribution is said to satisfy causal Markov property w.r.t. the DAG* $\mathcal{G}$ *if each variable is independent of its non-descendants given its parents.*

This is a common assumptions for the distribution induced by an SEM. With this assumption, one can deduce conditional independence between variables from the graph.

**Assumption 2** *(Causal Markov Property for multiple regimes). Given a set of DAGs* $(\mathcal{G}^u)_{u \in \{1,...,K\}}$ *and a set of joint distribution* $(p(\cdot | \mathcal{G}^u))_{u \in \{1,...,K\}}$, *we say that this set of distributions satisfies causal Markov property w.r.t. the set of DAGs* $(\mathcal{G}^u)_{u \in \{1,...,K\}}$ *if for every* $u$: $p(\cdot | \mathcal{G}^u)$ *satisfy causal Markov property w.r.t the DAG* $\mathcal{G}^u$.

**Definition 5** *(Causal Minimality, Gong et al. (2022)). Consider a distribution* $p$ *and a DAG* $\mathcal{G}$, *we say this distribution satisfies causal minimality w.r.t.* $\mathcal{G}$ *if it is Markovian w.r.t.* $\mathcal{G}$ *but not to any proper subgraph of* $\mathcal{G}$.

**Assumption 3** *(Causal Minimality for multiple regimes). Given a set of DAGs* $(\mathcal{G}^u)_{u \in \{1,...,K\}}$ *and a set of joint distribution* $(p(\cdot | \mathcal{G}^u))_{u \in \{1,...,K\}}$, *we say that this set of distributions satisfies causal minimality w.r.t. the set of DAGs* $(\mathcal{G}^u)_{u \in \{1,...,K\}}$ *if for every* $u$: $p(\cdot | \mathcal{G}^u)$ *satisfy causal minimality w.r.t the DAG* $\mathcal{G}^u$.

**Assumption 4** *(Causal Sufficiency). A set of observed variables* $V$ *is causally sufficient for a process* $\boldsymbol{X}_t$ *if and only if in the process every common cause of any two or more variables in* $\boldsymbol{V}$ *is in* $\boldsymbol{V}$ *or has the same value for all units in the population.*

This assumption implies there are no latent confounders present in the time-series data.

**Assumption 5** *(Well-defined Density). We assume the joint likelihood induced by the CASTOR SEM (Eq. (3)) is absolutely continuous w.r.t. a Lebesgue or counting measure and* $\forall u$ : $|\log p(\boldsymbol{X}_{t \in \mathcal{E}_u}; \mathcal{G}^u)| < \infty$ *for all possible* $\mathcal{G}^u$.

This presumption ensures that the resulting distribution possesses a well-defined probability density function. It is also necessary for Markov factorization property.

**Assumption 6** *(CASTOR in DAG space). We assume the CASTOR framework can only return the solutions from DAG space.*

**Definition 6** *(Source model [Gong et al. (2022)](#)) For any MTS defined as follows:*

$$X_t^i = f_{i,t}\left(\mathbf{Pa}_G^i(< t), \mathbf{Pa}_G^i(t)\right) + \epsilon_t^i \tag{20}$$

*A source model characterizes the initial conditions:*

$$X_s^i = f_{i,s}\left(\mathbf{Pa}_G^i\right) + \epsilon_s^i \tag{21}$$

*for $s \in [0, \mathcal{S}]$, where $\mathcal{S}$ is the length for the initial conditions and $\mathbf{Pa}_G^i$ contains the parents for node $i$. We define $p_s(\mathbf{X}_{0:\mathcal{S}})$ as the induced joint distribution for the initial conditions.*

**Assumption 7** *(CASTOR initial condition). We assume that the initial conditions are known and source model is identifiable.*

### C.1 PROOF OF THEOREM 1

Assuming the aforementioned assumptions we want to prove the theorem [1](#).

We consider $\mathcal{G} = (\mathcal{G}^u)_{u \in \{1,..,N_w\}}$, $\mathcal{E} = \cup_{u=1}^{N_w} \mathcal{E}_u$ where $N_w$ is the number of window, $\mathcal{G}^* = (\mathcal{G}^{*,u})_{u \in \{1,..,K\}}$, $K$ is the exact number of true regimes and $\mathcal{E}^* = \cup_{u=1}^K \mathcal{E}_u^*$. We denote $\mathcal{E}_c \mathcal{E}_\ell^*$ the set of time indexes that is shared between regime $c$ of our model estimation and the true regime $\ell$ and $q_\ell^* := \frac{|\mathcal{E}_\ell^*|}{T}, q_c := \frac{|\mathcal{E}_c|}{T}, q_{c\ell} := \frac{|\mathcal{E}_c \mathcal{E}_\ell^*|}{T}$. Our objective is to prove that for any estimation $(\hat{\mathcal{G}}, \hat{\mathcal{E}})$ : if $\exists u \in \{1, ..., K\}$ s.t. $\hat{\mathcal{G}}^u$ disagree with $\mathcal{G}^{*,u}$ on instantaneous or/and time lagged link, or any regime $\hat{\mathcal{E}}_u \in \hat{\mathcal{E}}$ is close to none of the true regimes in the sense of Kullback–Leibler divergence: $\mathcal{S}(\mathcal{G}^*, \mathcal{E}^*) > \mathcal{S}(\hat{\mathcal{G}}, \hat{\mathcal{E}})$.
We have by Eq [(9)](#)

$$\mathcal{S}(\mathcal{G}, \mathcal{E}) := \sup_{\theta, \mathcal{G}} \frac{1}{T} \sum_{u=1}^{N_u} \sum_{t \in \mathcal{E}_u} \log f^u(\mathbf{X}_t) - \lambda |\mathcal{G}^u|,$$

where $\lambda$ is the sparsity penalty coefficient and $f^u(\mathbf{X}_t) := \prod_{j=1}^d f_i^u\left(\mathbf{Pa}_{\mathcal{G}^u}^i(< t), \mathbf{Pa}_{\mathcal{G}^u}^i(t)\right)$ with $f_i^u\left(\mathbf{Pa}_{\mathcal{G}^u}^i(< t), \mathbf{Pa}_{\mathcal{G}^u}^i(t)\right)$ the function used to describe the distribution family in Eq [(7)](#).
We will structure the proof as follows:

- Prove that if the score is optimized, then all the estimated regimes will be pure (have only elements of the same true regime).

- Prove that, when the regimes are pure and $N_w = K$, we have $\mathcal{S}(\mathcal{G}^*, \mathcal{E}^*) > \mathcal{S}(\hat{\mathcal{G}}, \hat{\mathcal{E}})$ for any estimation $\hat{\mathcal{G}}$ where $\exists u \in \{1, ..., K\}$ s.t $\hat{\mathcal{G}}^u$ disagrees with $\mathcal{G}^{*,u}$ on instantaneous or/and time lagged link.

### C.1.1 OPTIMIZING THE SCORE WILL LEAD TO PURE REGIMES

Ignoring penalty terms, we have:

$$
\begin{aligned}
-\mathcal{S}(\mathcal{G}, \mathcal{E}) &= -\sup_{\theta} \sum_{c=1}^{N_w} \sum_{\ell=1}^{K} q_{c\ell} \frac{1}{|\mathcal{E}_e \mathcal{E}_\ell^*|} \sum_{t \in \mathcal{E}_e \mathcal{E}_\ell^*} [\log f^c(\boldsymbol{X}_t)] \\
&\to -\sup_{\phi} \sum_{c=1}^{N_w} \sum_{\ell=1}^{K} q_{c\ell} \mathop{\mathbb{E}}_{\boldsymbol{X}_t \sim p} [\log f^c] \\
&= -\sup_{\theta} \sum_{c=1}^{N_w} \sum_{\ell=1}^{K} q_{c\ell} \mathop{\mathbb{E}}_{\boldsymbol{X}_t \sim p} \left[ \sum_{i=1}^{d} \log f_i^c \left( \mathbf{Pa}_{\mathcal{G}^c}^i(<t), \mathbf{Pa}_{\mathcal{G}^c}^i(t) \right) \right] \\
&= -\sup_{\theta} \sum_{c=1}^{N_w} \sum_{\ell=1}^{K} \sum_{i=1}^{d} q_{c\ell} \\
&\quad \mathop{\mathbb{E}}_{\boldsymbol{X}_t \sim p} \left[ -\log \frac{p\left( X_t^i \mid \left( \mathbf{Pa}_{\mathcal{G}*,\ell}^i(<t), \mathbf{Pa}_{\mathcal{G}*,\ell}^i(t) \right) \right)}{f_i^c \left( \mathbf{Pa}_{\mathcal{G}^c}^i(<t), \mathbf{Pa}_{\mathcal{G}^c}^i(t) \right)} + \log p \left( X_t^i \mid \left( \mathbf{Pa}_{\mathcal{G}*,\ell}^i(<t), \mathbf{Pa}_{\mathcal{G}*,\ell}^i(t) \right) \right) \right] \\
&= -\sup_{\theta} \sum_{c=1}^{N_w} \sum_{\ell=1}^{K} \sum_{i=1}^{d} q_{c\ell} \\
&\quad \mathop{\mathbb{E}}_{\boldsymbol{X}_t \sim p} \Big[ -\mathrm{D}_{\mathrm{KL}} \left( p\left( X_t^i \mid \left( \mathbf{Pa}_{\mathcal{G}*,\ell}^i(<t), \mathbf{Pa}_{\mathcal{G}*,\ell}^i(t) \right) \right) \| f_i^c \left( \mathbf{Pa}_{\mathcal{G}^c}^i(<t), \mathbf{Pa}_{\mathcal{G}^c}^i(t) \right) \right) \\
&\quad - \mathrm{H} \left( p\left( X_t^i \mid \left( \mathbf{Pa}_{\mathcal{G}*,\ell}^i(<t), \mathbf{Pa}_{\mathcal{G}*,\ell}^i(t) \right) \right) \right) \Big] \\
&= -\sup_{\theta} \sum_{c=1}^{N_w} \sum_{\ell=1}^{K} \sum_{i=1}^{d} q_{c\ell} \mathop{\mathbb{E}}_{\boldsymbol{X}_t \sim p} \left[ -\mathrm{D}_{\mathrm{KL}} \left( p\left( X_t^i \mid \left( \mathbf{Pa}_{\mathcal{G}*,\ell}^i(<t), \mathbf{Pa}_{\mathcal{G}*,\ell}^i(t) \right) \right) \| f_i^c \left( \mathbf{Pa}_{\mathcal{G}^c}^i(<t), \mathbf{Pa}_{\mathcal{G}^c}^i(t) \right) \right) \right] \\
&\quad - \sup_{\theta} \sum_{c=1}^{N_c} \sum_{\ell=1}^{K} \sum_{i=1}^{d} q_{c\ell} \mathop{\mathbb{E}}_{\boldsymbol{X}_t \sim p} \left[ -\mathrm{H} \left( p\left( X_t^i \mid \left( \mathbf{Pa}_{\mathcal{G}*,\ell}^i(<t), \mathbf{Pa}_{\mathcal{G}*,\ell}^i(t) \right) \right) \right) \right] \\
&= \inf_{\theta} \sum_{c=1}^{N_w} \sum_{\ell=1}^{K} \sum_{j=1}^{d} q_{c\ell} \mathop{\mathbb{E}}_{\boldsymbol{X}_t \sim p} \left[ \mathrm{D}_{\mathrm{KL}} \left( p\left( X_t^i \mid \left( \mathbf{Pa}_{\mathcal{G}*,\ell}^i(<t), \mathbf{Pa}_{\mathcal{G}*,\ell}^i(t) \right) \right) \| f_i^c \left( \mathbf{Pa}_{\mathcal{G}^c}^i(<t), \mathbf{Pa}_{\mathcal{G}^c}^i(t) \right) \right) \right] \\
&\quad + \sum_{c=1}^{N_w} \sum_{\ell=1}^{K} \sum_{j=1}^{d} q_{c\ell} \mathop{\mathbb{E}}_{\boldsymbol{X}_t \sim p} \left[ \mathrm{H} \left( p\left( X_t^i \mid \left( \mathbf{Pa}_{\mathcal{G}*,\ell}^i(<t), \mathbf{Pa}_{\mathcal{G}*,\ell}^i(t) \right) \right) \right) \right] \\
&= \inf_{\theta} \sum_{c=1}^{N_w} \sum_{\ell=1}^{K} \sum_{i=1}^{d} q_{c\ell} \mathop{\mathbb{E}}_{\boldsymbol{X}_t \sim p} \left[ \mathrm{D}_{\mathrm{KL}} \left( p\left( X_t^i \mid \left( \mathbf{Pa}_{\mathcal{G}*,\ell}^i(<t), \mathbf{Pa}_{\mathcal{G}*,\ell}^i(t) \right) \right) \| f_i^c \left( \mathbf{Pa}_{\mathcal{G}^c}^i(<t), \mathbf{Pa}_{\mathcal{G}^c}^i(t) \right) \right) \right] \\
&\quad + \sum_{\ell=1}^{K} \sum_{i=1}^{d} \left( \sum_{c=1}^{N_c} q_{c\ell} \right) \mathop{\mathbb{E}}_{\boldsymbol{X}_t \sim p} \left[ \mathrm{H} \left( p\left( X_t^i \mid \left( \mathbf{Pa}_{\mathcal{G}*,\ell}^i(<t), \mathbf{Pa}_{\mathcal{G}*,\ell}^i(t) \right) \right) \right) \right] \\
&= \inf_{\theta} \sum_{c=1}^{N_w} \sum_{\ell=1}^{K} \sum_{i=1}^{d} q_{c\ell} \mathop{\mathbb{E}}_{\boldsymbol{X}_t \sim p} \left[ \mathrm{D}_{\mathrm{KL}} \left( p\left( X_t^i \mid \left( \mathbf{Pa}_{\mathcal{G}*,\ell}^i(<t), \mathbf{Pa}_{\mathcal{G}*,\ell}^i(t) \right) \right) \| f_i^c \left( \mathbf{Pa}_{\mathcal{G}^c}^i(<t), \mathbf{Pa}_{\mathcal{G}^c}^i(t) \right) \right) \right] \\
&\quad + \sum_{\ell=1}^{K} \sum_{i=1}^{d} q_\ell^* \mathop{\mathbb{E}}_{\boldsymbol{X}_t \sim p} \left[ \mathrm{H} \left( p\left( X_t^i \mid \left( \mathbf{Pa}_{\mathcal{G}*,\ell}^i(<t), \mathbf{Pa}_{\mathcal{G}*,\ell}^i(t) \right) \right) \right) \right]
\end{aligned}
\tag{22}
$$

Note that $\theta$ could be the parameters of the neural networks used in Eq (15) for non linear causal relationship or $\theta = (\mathcal{G}^u)_{u \in \{1,..,N_w\}}$ for linear case Eq 10.

For the score of ground truth (ignoring penalty terms):

$$
-\mathcal{S}\left(\mathcal{G}^*, \mathcal{E}^*\right) \to 0 + \sum_{\ell=1}^{K} \sum_{i=1}^{d} q_\ell^* \mathop{\mathbb{E}}_{\boldsymbol{X}_t \sim p} \left[ \mathrm{H} \left( p\left( X_t^i \mid \left( \mathbf{Pa}_{\mathcal{G}*,\ell}^i(<t), \mathbf{Pa}_{\mathcal{G}*,\ell}^i(t) \right) \right) \right) \right] \text{ ( by Assumption 5 5)} \tag{23}
$$

Combining Equation (22) and Equation (23) , we have (considering penalty terms):

$$
\mathcal{S}\left(\mathcal{G}^*, \mathcal{E}^*\right) - \mathcal{S}(\mathcal{G}, \mathcal{E}) = \inf_\theta \sum_{c=1}^{N_w} \sum_{\ell=1}^{K} \sum_{i=1}^{d} q_{c\ell}
$$
$$
\mathop{\mathbb{E}}_{\boldsymbol{X}_t \sim p} \left[ D_{\mathrm{KL}} \left( p\left( X_t^i \mid \left( \mathbf{Pa}_{\mathcal{G}^*, \ell}^i(<t), \mathbf{Pa}_{\mathcal{G}^*, \ell}^i(t) \right) \right) \| f_i^c \left( \mathbf{Pa}_{\mathcal{G}^c}^i(<t), \mathbf{Pa}_{\mathcal{G}^c}^i(t) \right) \right) \right] \quad (24)
$$
$$
+ \lambda \left( \sum_{c=1}^{N_w} |\mathcal{G}^c| - \sum_{\ell=1}^{K} |\mathcal{G}^{*, \ell}| \right)
$$

The first term in Equation (24) is the score term, others are penalty term.

In the following lines, our goal is to demonstrate that optimizing the score term ensures that all identified regimes will accurately match the real regimes. In other words, each estimated regime will be a true representation of an actual one. Additionally, by shifting samples from less significant regimes (regimes with few samples) to the most similar significant regimes, our variable $N_w$ will eventually stabilize at the value of K. To do this, we will proceed by contradiction:

Suppose the score term in Eq (24) is optimized and there exists a regime $e$ that is **not pure**, i.e., there exist $a, b \in [K]$ with $a \neq b$ but $q_{ea} > 0$ and $q_{eb} > 0$. Since they are different distributions for two different regimes with two different causal graphs, there exists $i \in \{1, ..., d\}$ such that $p\left( X_t^i \mid \left( \mathbf{Pa}_{\mathcal{G}^*, a}^i(<t), \mathbf{Pa}_{\mathcal{G}^*, a}^i(t) \right) \right) \neq p\left( X_t^i \mid \left( \mathbf{Pa}_{\mathcal{G}^*, b}^i(<t), \mathbf{Pa}_{\mathcal{G}^*, b}^i(t) \right) \right)$. Then the score term in Equation (24) has the following lower bound:

$$
\inf_\theta \sum_{\ell=1}^{K} \sum_{i=1}^{d} q_{e\ell} \mathop{\mathbb{E}}_{\boldsymbol{X}_t \sim p} \left[ D_{\mathrm{KL}} \left( p\left( X_t^i \mid \left( \mathbf{Pa}_{\mathcal{G}^*, \ell}^i(<t), \mathbf{Pa}_{\mathcal{G}^*, \ell}^i(t) \right) \right) \| f_i^e \left( \mathbf{Pa}_{\mathcal{G}^e}^i(<t), \mathbf{Pa}_{\mathcal{G}^e}^i(t) \right) \right) \right]
$$
$$
\geq \inf_\theta \left\{ q_{ea} \mathop{\mathbb{E}}_{\boldsymbol{X}_t \sim p} \left[ D_{\mathrm{KL}} \left( p\left( X_t^i \mid \left( \mathbf{Pa}_{\mathcal{G}^*, a}^i(<t), \mathbf{Pa}_{\mathcal{G}^*, a}^i(t) \right) \right) \| f_i^e \left( \mathbf{Pa}_{\mathcal{G}^e}^i(<t), \mathbf{Pa}_{\mathcal{G}^e}^i(t) \right) \right) \right] \right. \quad (25)
$$
$$
\left. + q_{eb} \mathop{\mathbb{E}}_{\boldsymbol{X}_t \sim p} \left[ D_{\mathrm{KL}} \left( p\left( X_t^i \mid \left( \mathbf{Pa}_{\mathcal{G}^*, b}^i(<t), \mathbf{Pa}_{\mathcal{G}^*, b}^i(t) \right) \right) \| f_i^e \left( \mathbf{Pa}_{\mathcal{G}^e}^i(<t), \mathbf{Pa}_{\mathcal{G}^e}^i(t) \right) \right) \right] \right\}
$$

As we assumed that the score term in Eq (24) is optimized, it means that:

$$
0 = \inf_\theta \sum_{c=1}^{N_w} \sum_{\ell=1}^{K} \sum_{i=1}^{d} q_{c\ell} \mathop{\mathbb{E}}_{\boldsymbol{X}_t \sim p} \left[ D_{\mathrm{KL}} \left( p\left( X_t^i \mid \left( \mathbf{Pa}_{\mathcal{G}^*, \ell}^i(<t), \mathbf{Pa}_{\mathcal{G}^*, \ell}^i(t) \right) \right) \| f_i^c \left( \mathbf{Pa}_{\mathcal{G}^c}^i(<t), \mathbf{Pa}_{\mathcal{G}^c}^i(t) \right) \right) \right]
$$
$$
\Rightarrow 0 = \inf_\theta \sum_{\ell=1}^{K} \sum_{i=1}^{d} q_{e\ell} \mathop{\mathbb{E}}_{\boldsymbol{X}_t \sim p} \left[ D_{\mathrm{KL}} \left( p\left( X_t^i \mid \left( \mathbf{Pa}_{\mathcal{G}^*, \ell}^i(<t), \mathbf{Pa}_{\mathcal{G}^*, \ell}^i(t) \right) \right) \| f_i^e \left( \mathbf{Pa}_{\mathcal{G}^e}^i(<t), \mathbf{Pa}_{\mathcal{G}^e}^i(t) \right) \right) \right] \quad (26)
$$
$$
\Rightarrow \left\{ \begin{array}{l} D_{\mathrm{KL}} \left( p\left( X_t^i \mid \left( \mathbf{Pa}_{\mathcal{G}^*, a}^i(<t), \mathbf{Pa}_{\mathcal{G}^*, a}^i(t) \right) \right) \| f_i^e \left( \mathbf{Pa}_{\mathcal{G}^e}^i(<t), \mathbf{Pa}_{\mathcal{G}^e}^i(t) \right) \right) = 0 \\ D_{\mathrm{KL}} \left( p\left( X_t^i \mid \left( \mathbf{Pa}_{\mathcal{G}^*, b}^i(<t), \mathbf{Pa}_{\mathcal{G}^*, b}^i(t) \right) \right) \| f_i^e \left( \mathbf{Pa}_{\mathcal{G}^e}^i(<t), \mathbf{Pa}_{\mathcal{G}^e}^i(t) \right) \right) = 0 \end{array} \right.
$$
$$
\Rightarrow \forall i \in \{1, ..., d\} : p\left( X_t^i \mid \left( \mathbf{Pa}_{\mathcal{G}^*, a}^i(<t), \mathbf{Pa}_{\mathcal{G}^*, a}^i(t) \right) \right) = p\left( X_t^i \mid \left( \mathbf{Pa}_{\mathcal{G}^*, b}^i(<t), \mathbf{Pa}_{\mathcal{G}^*, b}^i(t) \right) \right)
$$

and the last line, Eq (26), is a contradiction because the two distributions represent two different regimes with two different graphs. Hence, if the score term of Eq (24) is optimized all the estimated regimes will be pure.

**First case:** If we matched the samples of less significant regimes to the wrong regimes, the regime is not pure and then the score term is not optimized (contradiction).

**Second case:** If we eliminate a lot of regimes such that $N_w \leq K - 1$, at least one of our estimated regimes will not be pure and this contradicts the assumption of optimized score term (same reasoning).

Based on this reasoning, optimizing the score term of Equation 24 will ensure convergence to the true number of regimes and also every regime will be pure.

## C.1.2 In case of edge disagreement $\mathcal{S}\left(\mathcal{G}^*, \mathcal{E}^*\right) > \mathcal{S}(\hat{\mathcal{G}}, \hat{\mathcal{E}})$

Now we will show that Eq (24 )is positive, if $\exists u \in \{1, ..., K\}$ s.t $\hat{\mathcal{G}}^u$ disagrees with $\mathcal{G}^{*, u}$ on instantaneous or/and time lagged link.

To simplify the notation, we denote by $p^{(u)}$ the distribution $p(. | \mathcal{G}^u)$ the optimal distribution that describes the cgm of regime $u$. We assume that each estimated regime $\hat{\mathcal{E}}_c$ ($c \in \{1, \ldots, N_w\}, N_w \geq K$) contains samples from same true regime. Then Equation (24) has lower bound:

$$
\inf_\theta \sum_{\ell=1}^{K} q_\ell^* \mathop{\mathbb{E}}_{\boldsymbol{X}_t \sim p} D_{\mathrm{KL}} \left( p^{(\ell)} \| f^\ell \right)
$$
$$
\geq (\min_\ell q_\ell^*) \inf_\theta \sum_{\ell=1}^{K} D_{\mathrm{KL}} \left( p^{(\ell)} \| f^\ell \right) \quad (27)
$$

Equation (27) is positive if and only if $\eta(\mathcal{G})$ is positive.

$$\eta(\mathcal{G}) := \inf_{\theta} \sum_{\ell=1}^{K} D_{KL}\left(p^{(\ell)} \| f^{\ell}\right) \tag{28}$$

Let assume that $\exists r \in\in \{1, ..., K\}$ s.t $\hat{\mathcal{G}}^r$ disagrees with $\mathcal{G}^{*,r}$ on instantaneous or/and time lagged link. We follow the same intuition as Gong et al. (2022); Peters et al. (2013; 2017):

$$
\begin{aligned}
&D_{KL}\left(p^{(r)} \| f^r\right) \\
=&D_{KL}\left[p_s^{(r)} \| f_s^r\right] + \sum_{t=\mathcal{S}+1}^{|\mathcal{E}_r^*|} \mathbb{E}_{p^{(r)}(\boldsymbol{X}_{0:t-1})}\left[D_{KL}\left[p^{(r)}\left(\boldsymbol{X}_t \mid \boldsymbol{X}_{0:t-1}\right) \| f^r\left(\boldsymbol{X}_t \mid \boldsymbol{X}_{0:t-1}\right)\right]\right]
\end{aligned}
\tag{29}
$$

Based on Assumption 7, we know that the source model is known, which leads to the following result:

$$
\text{Eq (29)} \iff D_{KL}\left(p^{(r)} \| f^r\right) = \sum_{t=\mathcal{S}+1}^{|\mathcal{E}_r^*|} \mathbb{E}_{p^{(r)}(\boldsymbol{X}_{0:t-1})}\left[D_{KL}\left[p^{(r)}\left(\boldsymbol{X}_t \mid \boldsymbol{X}_{0:t-1}\right) \| f^r\left(\boldsymbol{X}_t \mid \boldsymbol{X}_{0:t-1}\right)\right]\right]
\tag{30}
$$

We will show that $D_{KL}\left(p^{(r)} \| f^r\right)$ is positive in two cases:

- **Disagreement on lagged parents only.** This means that for all $t \in [\mathcal{S}+1, T]$, the instantaneous connections at $t$ for $\hat{\mathcal{G}}^r$ and $\mathcal{G}^{*,r}$ are the same, and $\exists t \in [\mathcal{S}+1, T]$ and $i \in \{1, ..., d\}$ such that $\mathbf{Pa}_{\mathcal{G}^{*,r}}^{X_t^i}(< t) \neq \overline{\mathbf{Pa}}_{\hat{\mathcal{G}}^r}^{X_t^i}(< t)$. We can use a similar argument as the theorem 1 in Peters et al. (2013). Without loss of generality, we assume under $\hat{\mathcal{G}}^r$, we have $X_{t-\tau}^j \to X_t^i$ and there is no connections between them under $\mathcal{G}^{*,r}$. Thus, from Markov conditions, we have

$$X_t^i \perp\!\!\!\perp X_{t-\tau}^j \mid \boldsymbol{X}_{0:t-1} \cup \mathrm{ND}_t^{X^i} \setminus \left\{X_t^i, X_{t-\tau}^j\right\}$$

  under $\mathcal{G}^{*,r}$, where $\mathrm{ND}_t^{X^i}$ are the non-descendants of node $X_t^i$ at some time $t$. However, from the causal minimality and Proposition 6.16 in Peters et al. (2017), we have

$$X_t^i \not\perp\!\!\!\perp X_{t-\tau}^j \mid \boldsymbol{X}_{0:t-1} \cup \mathrm{ND}_t^{X^i} \setminus \left\{X_t^i, X_{t-\tau}^j\right\}$$

  under $\hat{\mathcal{G}}^r$, and we have $D_{KL}\left[p^{(r)}\left(\boldsymbol{X}_t \mid \boldsymbol{X}_{0:t-1}\right) \| f^r\left(\boldsymbol{X}_t \mid \boldsymbol{X}_{0:t-1}\right)\right] \neq 0$. Hence, $D_{KL}\left(p^{(r)} \| f^r\right) \neq 0$

- **Disagreement on instantaneous parents.** In this Section we will use two different results one for the linear and the other one for the non linear case.

  - *Linear case.* For this case, we will use Theorem 1 in Peters & Bühlmann (2014). In this theorem, the author confirms that the graph is identifiable for linear models with Gaussian additive noise, if for each $j \in \{1, \ldots, d\}$, the weights of the causal relations $\beta_{jk} \neq 0$ for all $k \in \mathbf{PA}_j^{\mathcal{G}_0}$. For our instantaneous links, we have all the weights of the parents are non null. Hence, the instantaneous links are identifiable. Otherwise if $D_{KL}\left(p^{(r)} \| f^r\right) \neq 0$

  - *Non linear case.* Using Theorem 2 from Peters et al. (2012), we can notice that our instantaneous links are Identifiable Functional Model Class, $(\mathcal{B}, \mathcal{F})$-IFMOC, they belongs exactly to the 3rd class of Lemma 3 in Peters et al. (2012): nonlinear ANMs: $\mathcal{F}_3 = \{f(X, n) = \phi(X) + n\}$ $\mathcal{B}_3 = \{(\phi, X, N) \text{ not lin., Gauss, Gauss }\} \setminus \tilde{B}_3$. Hence our instantaneous links are identifiable, otherwise, $D_{KL}\left(p^{(r)} \| f^r\right) \neq 0$.

Based on the above reasoning, we can show that if $\exists r \in\in \{1, ..., K\}$ s.t., $\hat{\mathcal{G}}^r$ disagree with $\mathcal{G}^{*,r}$ on instantaneous or/and time lagged links, $D_{KL}\left(p^{(r)} \| f^r\right) \neq 0$.

Thus, $\eta(\mathcal{G}) > 0$. Then as we assume in Theorem 1 that $\lambda$ is sufficient small would implies Equation (26) is positive.

If $|\hat{\mathcal{G}}^r| \geq |\mathcal{G}^{*,r}|$ then clearly Eq 26 is positive. Let $\mathbb{G}^+ := \left\{\hat{\mathcal{G}}^r \in \mathbb{G} | |\hat{\mathcal{G}}^r| < |\mathcal{G}^{*,r}|\right\}$. To make sure that we have $\mathcal{S}\left(\mathcal{G}^*, \mathcal{E}^*\right) - \mathcal{S}(\mathcal{G}, \mathcal{E}) > 0$ for all $\mathcal{G} \in \mathbb{G}^+$, we need to pick $\lambda$ sufficiently small. Choosing $0 < \lambda < \min_{\mathcal{G} \in \mathbb{G}^+} \frac{\eta(\mathcal{G})}{\left(\sum_{c=1}^{N_w} |\mathcal{G}^c| - \sum_{\ell=1}^{K} |\mathcal{G}^{*,\ell}|\right)}$ is sufficient.

# D  FURTHER EXPERIMENTAL RESULTS

## D.1  SYNTHETIC DATA

We employ the Erdos–Rényi (ER) (Newman, 2018) model with mean degrees of 1 or 2 to generate lagged graphs, and the Barabasi–Albert (BA) (Barabási & Albert, 1999) model with mean degrees 4 for instantaneous graphs. The maximum number of lags, $L$, is set at 1 or 2. We experiment with varying numbers of nodes $\{5, 10, 20, 40\}$ and different numbers of regimes $\{2, 3, 4, 5\}$, each representing diverse causal graphs. The length of each regime is randomly sampled from the set $\{300, 400, 500, 600\}$.

- **Linear case.** Data is generated as follows:

$$\forall u \in \{1, ..., K\}, \forall t \in \mathcal{E}_u : \boldsymbol{X}_t = \boldsymbol{X}_t \boldsymbol{G}_0^u + \sum_{\tau=1}^{L} \boldsymbol{X}_{t-\tau} \boldsymbol{G}_\tau^u + \boldsymbol{\epsilon}_t,$$

  with $\boldsymbol{G}_0^u$ is adjacency matrix of the generated graph by BA model, $\forall \tau \in \{1, .., L\} : \boldsymbol{G}_\tau^u$ are the adjacency of the time lagged graphs generated by ER and $\epsilon_t \sim \mathcal{N}(0, I)$, follows to a normal distribution.

- **Non-linear case.** The formulation used to generated the data is:

$$\forall u \in \{1, ..., K\}, \forall t \in \mathcal{E}_u : X_t^i = g_i^u \left( \mathbf{Pa}_{\mathcal{G}^u}^i(<t), \mathbf{Pa}_{\mathcal{G}^u}^i(t) \right) + \epsilon_t^i,$$

  where $g_i^u$ is a general differentiable linear/non-linear function and $\epsilon_t^i \sim \mathcal{N}(0, 1)$, follows a normal distribution. The function $g_i^u$ is a random combination between a linear transformation and a randomly chosen function from the set: $\{\texttt{Tanh, LeakyReLU, ReLU}\}$.

## D.2  BASELINES

All used benchmarks for the synthetic experiments are run by using publicly available libraries: VARLINGAM (Hyvärinen et al., 2010) is implemenented in the $\texttt{lingam}$[4] python package. RPCMCI (Saggioro et al., 2020) is implemented in $\texttt{Tigramite}$[5] and DYNOTEARS (Pamfil et al., 2020) on $\texttt{causalnex}$[6] package. We fine tuned the parameters to achieve the optimal graph for each model.

For CASTOR, an edge threshold of 0.4 is selected. In the linear scenario, we establish $\zeta = 100$ as the minimum regime duration, while in the non-linear context, $\zeta$ is set at 200. To demonstrate the model's robustness to the choice of the window size, we train CASTOR using diverse window sizes, specifically $w = 200$ or $w = 300$. For the sparsity coefficient, we use $\lambda = 0.05$. In order to optimise our M-step, we use L-BFGS-B algorithm (Zhu et al., 1997).

## D.3  NESTIM BRAIN CONNECTIVITY

| Model | F1 FMRI | Reg Acc |
|---|---|---|
| PWGC | **66.7** | - |
| VARLINGAM | 47.6 | - |
| CASTOR | **66.5** | **80** |

Table 3: F1 Scores for FMRI data, 'Reg' indicates regime and 'Reg Acc' refers to regime identification accuracy.

Finally we test the efficacy of CASTOR on FMRI imaging dataset Nestim (Smith et al., 2011), a resource commonly utilized as a benchmark in the field of temporal causal discovery (Löwe et al. (2022); Khanna & Tan (2019); Assaad et al. (2022)). Each time series in the dataset simulates the neural signals for an individual human subject, encompassing $d = 5$ distinct nodes. Notably, the majority of time series in the Nestim dataset have the same summary causal graph. In order to create

---

[4] https://lingam.readthedocs.io/en/latest/
[5] https://jakobrunge.github.io/tigramite/
[6] https://causalnex.readthedocs.io/en/latest/

MTS with multiple regimes, we generated additional synthetic data using the generative process outlined in the previous section 5.1. Our choice of baseline models aligns with those utilized in the IT monitoring data evaluation, as detailed in Section 6.2. Table 3 shows that CASTOR successfully infers exact number of regimes and it succeeds to detect 80% of the real FMRI data. Even if the regime accuracy is 80%, CASTOR shows robustness and versatility in identifying complex temporal relationships within this type of data by achieving similar results to the baselines that have access to the true regime indices.

### D.4 FURTHER EXPERIMENTS: LINEAR CASE

| | | | $K = 2$ | | $K = 3$ | | $K = 4$ | |
|---|---|---|---|---|---|---|---|---|
| $d$ | Method | Type | SHD | F1 | SHD | F1 | SHD | F1 |
| 5 | VARLINGAM | Inst. | 7 | $22.9_{\pm5.6}$ | 13 | $18.4_{\pm3.6}$ | 14 | $24.1_{\pm6.5}$ |
| | | Lag | 9 | $8.01_{\pm12}$ | 14 | $15.9_{\pm4.2}$ | 14 | $8.20_{\pm6.5}$ |
| | RPCMCI | Inst. | - | - | - | - | - | - |
| | | Lag | - | - | - | - | - | - |
| | CASTOR | Inst. | **0** | $\mathbf{100}_{\pm0.0}$ | **0** | $\mathbf{100}_{\pm0.0}$ | **2** | $\mathbf{97.3}_{\pm1.9}$ |
| | | Lag | **0** | $\mathbf{100}_{\pm0.0}$ | **0** | $\mathbf{100}_{\pm0.0}$ | **1** | $\mathbf{97.2}_{\pm2.0}$ |
| 20 | VARLINGAM | Inst | 62 | $8.83 \pm 1.7$ | 90 | $10.3 \pm 2.4$ | 121 | $14.0 \pm 2.4$ |
| | | Lag | 48 | $2.96 \pm 0.2$ | 61 | $1.66 \pm 1.24$ | 93 | $2.30 \pm 0.7$ |
| | RPCMCI | Inst | - | - | - | - | - | - |
| | | Lag | 67 | $46.6 \pm 9.14$ | 178 | $14.1_{\pm2.3}$ | - | - |
| | CASTOR | Inst | **3** | $\mathbf{97.0}_{\pm2.7}$ | **28** | $\mathbf{88.1}_{\pm6.2}$ | 4 | $\mathbf{98.3}_{\pm1.4}$ |
| | | Lag | **0** | $\mathbf{100}_{\pm0}$ | 9 | $\mathbf{89.9}_{\pm}5.1$ | 1 | $\mathbf{99.7}_{\pm0.4}$ |

Table 4: F1 and SHD Scores by Models and Settings: $d$ indicates number of nodes, $K$ refers to the number of regimes, and 'Type' refers to the causal links as either instantaneous or time-lagged.

| | | | $K = 5$ | |
|---|---|---|---|---|
| $d$ | Method | Type | SHD | F1 |
| 10 | VARLINGAM | Inst. | 59 | $11.7_{\pm2.3}$ |
| | | Lag | 51 | $4.69_{\pm2.4}$ |
| | RPCMCI | Inst. | - | - |
| | | Lag | - | - |
| | CASTOR | Inst. | **4** | $\mathbf{98.4}_{\pm1.4}$ |
| | | Lag | **1** | $\mathbf{97.1}_{\pm1.4}$ |

Table 5: F1 and SHD Scores by Models and Settings: $d$ indicates number of nodes, $K$ refers to the number of regimes, and 'Type' refers to the causal links as either instantaneous or time-lagged.

In this section, we present additional results using linear synthetic data with varying numbers of nodes, specifically $\{5, 20\}$, and diverse numbers of regimes $\{2, 3, 4, 5\}$. Our model's performance is compared against RPCMCI (Saggioro et al., 2020) and VARLINGAM (Hyvärinen et al., 2010), and the results are displayed in Table 4, 5. CASTOR not only outperforms RPCMCI but consistently converges, accurately identifying the number of regimes and recovering both the regime indices and their respective underlying causal graphs. In scenarios with 5 different regimes, where RPCMCI fails to converge, CASTOR infers the true number of regimes, their partitions, and efficiently learns the causal graphs, achieving more than a 98% F1 score. Furthermore, in comparison to VARLINGAM (a method capable of modeling both lagged and instantaneous links in MTS data), CASTOR demonstrates markedly superior performance.

Figure 4 displays a comparison between the graphs estimated by CASTOR and the ground truth graphs. CASTOR effectively learns the causal graphs and the distinct regime partitions. In figure 5 we test CASTOR on data with $L = 2$ lags, hence CASTOR needs to estimate 3 adjacency matrices

in every regimes (the matrix for instantaneous links, first time lagged links and second time lagged links). We can notice that CASTOR estimates well the graphs and infers the regime partitions.

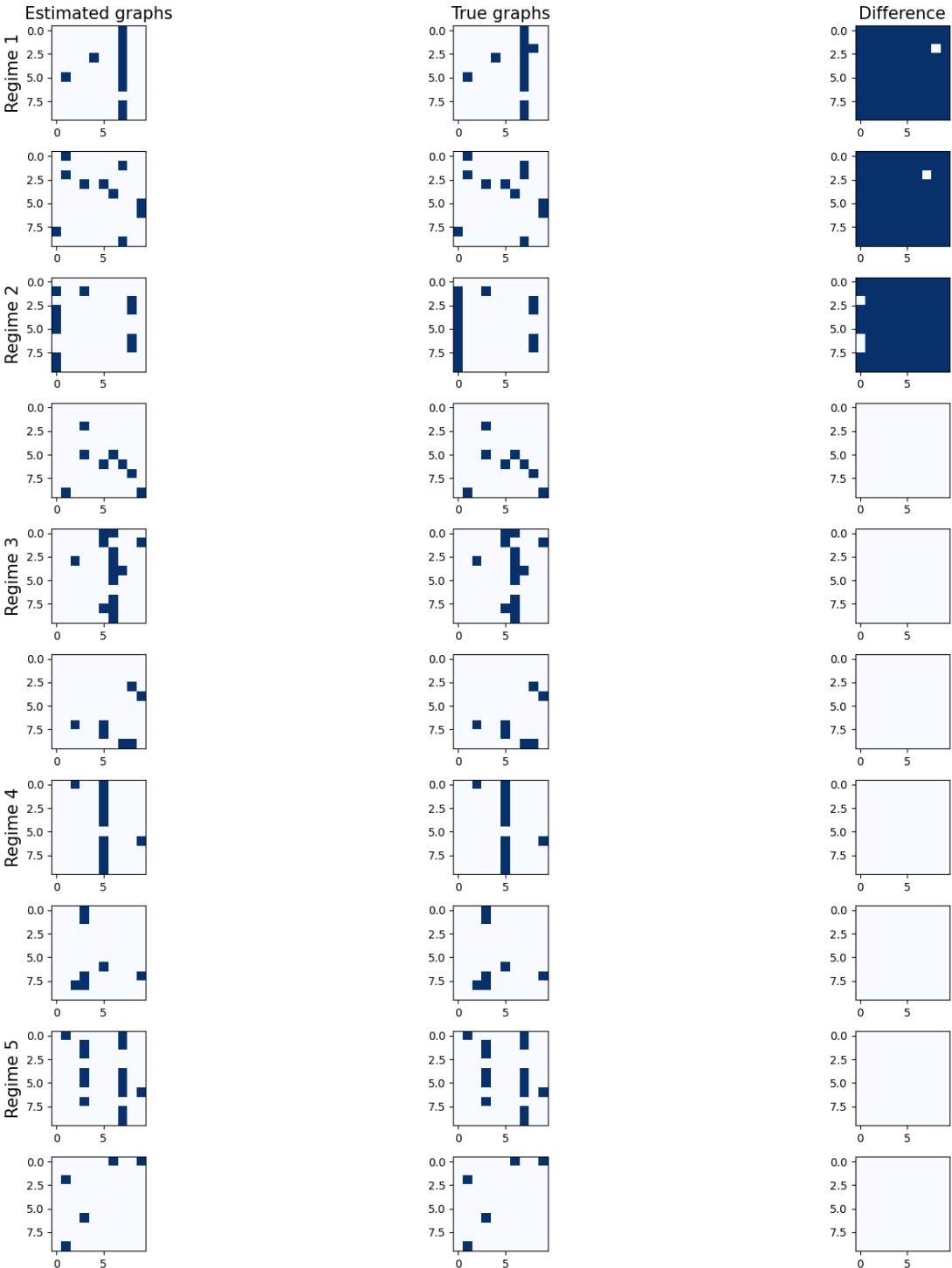

Figure 4: The estimated temporal causal graphs for five regimes (**Linear case**), with one matrix representing instantaneous links and another delineating time-lagged relations. The second column showcases the actual causal graphs, while the final column highlights the discrepancies between the estimated and true graphs.

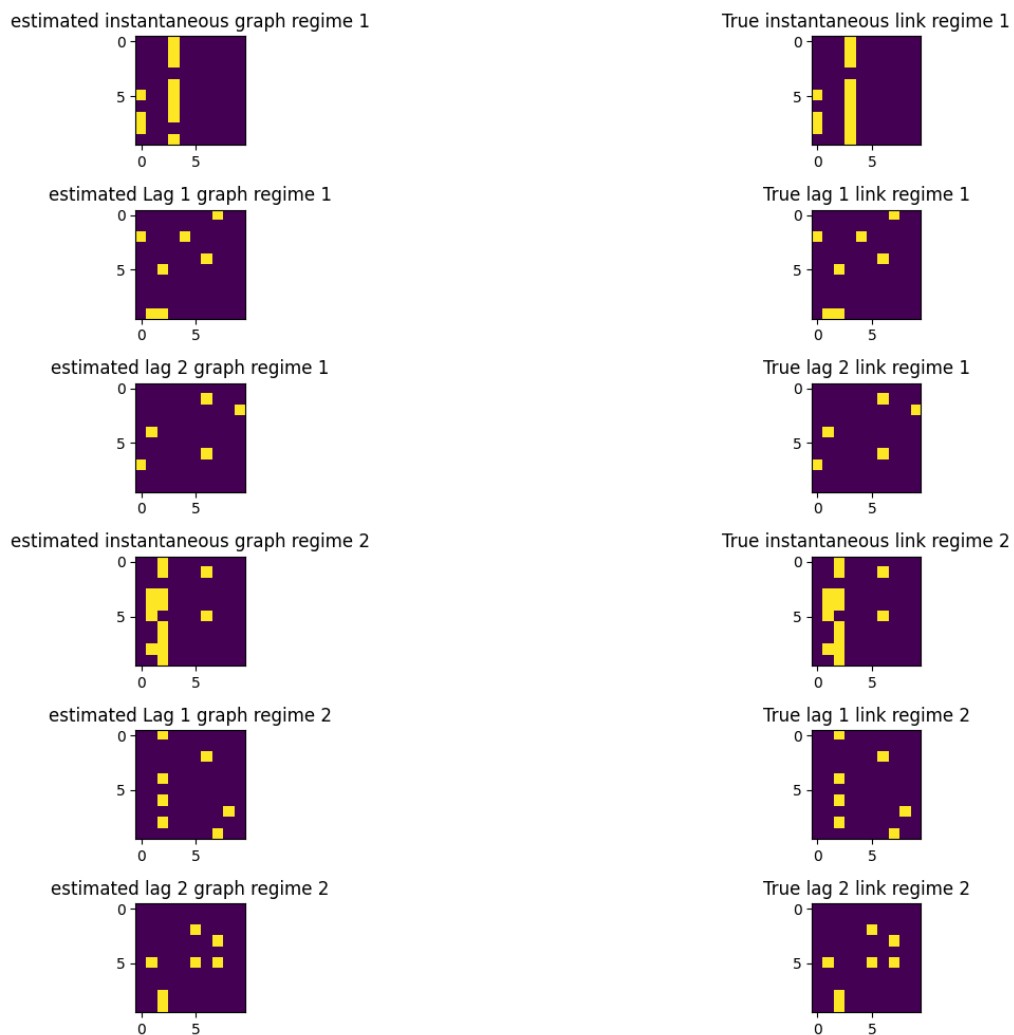

Figure 5: The estimated temporal causal graphs For MTS with 2 lags and 10 nodes and 2 regimes for the **linear case**

### D.5 FURTHER EXPERIMENTS: COMPARISON WITH CD-NOD AND RHINO

We conducted a comparative analysis with CD-NOD, a causal discovery model specifically designed for heterogeneous data and non-stationary time series (Huang et al., 2020). In the context of MTS with multiple regimes, CD-NOD learns a summary causal graph encapsulating the entire MTS.

It's worth noting that in the work by Huang et al. (2020), CD-NOD demonstrates the capability to learn both the regime partition and the summary causal graph. However, in our specific scenario, we utilized the publicly available version of CD-NOD within the `causal-learn`[7]. In this package, CD-NOD focuses on learning the summary causal graph while necessitating the availability of regime indexes.

Our experimental setup involves linear causal relations and diverse configurations, including 2, 3, and 4 regimes, each with varying numbers of nodes (10, 20, 40). For independence test of CD-NOD, we chose Fisher's Z conditional independence test for faster runs; note that we tested KCI CD-NOD, the model takes over 2000 seconds for 2 regimes, with F1 scores in a similar range. We systematically compared the performance of CD-NOD against CASTOR, with the evaluation centered on the summary causal graphs as the basis for comparison.

**Definition 7** *(Summary causal graph, (Assaad et al., 2022)) Let $(X_t)_{t \in \mathcal{T}}$ be a MTS and $\mathcal{G} = (V, E)$ the associated summary causal graph. The set of vertices in that graph consists of the set of components $X^1, \ldots, X^d$ at each time $t \in \mathbb{N}$. The edges $E$ of the graph are defined as follows: variables $X^p$ and $X^q$ are connected if and only if there exists some time $t$ and some time lag $\tau$ such that $X_{t-\tau}^p$ causes $X_t^q$ at time $t$ with a time lag of $0 \leq \tau$ for $p \neq q$ and with a time lag of $0 < \tau$ for $p = q$.*

| $d$ | Method | $K = 2$ | $K = 3$ | $K = 4$ |
|---|---|---|---|---|
| 10 | CD-NOD | 20.2 | 11.4 | 38.8 |
|    | CASTOR | **100** | **100** | **97.9** |
| 20 | CD-NOD | 25.2 | 23.7 | 12.7 |
|    | CASTOR | **100** | **97.2** | **93.4** |
| 40 | CD-NOD | 0 | 11.3 | 5.57 |
|    | CASTOR | **100** | **99.8** | **99.2** |

Table 6: F1 Scores by Models and Settings: $d$ indicates number of nodes and $K$ refers to the number of regimes. The comparison is made for linear relations

From Table 6, it is evident that despite having access to regime indexes, CD-NOD does not manage to outperform CASTOR in various settings (with an F1 score that does not exceed 26%). Additionally, a clear trend emerges where CD-NOD's performance declines when the number of nodes is 40. On the contrary, CASTOR exhibits consistent performance across different settings, achieving a F1 score of over 93% in all scenarios.

We conducted a thorough comparison between CASTOR and Rhino, a cutting-edge model in causal discovery. Rhino is specifically designed to infer causal graphs from time series data, particularly Multivariate Time Series (MTS) characterized by history-dependent noise. This means that Rhino considers the possibility of non-stationary noise, where its distribution is influenced by past observations. The work by Gong et al. (2022) involves optimizing a variational model to discern the inherent graph structure within the data.

It's worth noting that Rhino's training involves at least 50 MTS, all sharing the same causal graph, to learn the target Directed Acyclic Graph (DAG). In our comparison with Rhino, we generated diverse MTS compositions with distinct regimes. We ensured that each regime contained a minimum of 10,000 data points (calculated as $50 \times 200$, where 50 is the number of MTS, and 200 is the length of each MTS). This approach aimed to replicate a scenario similar to the one presented in the referenced paper.

---

[7] https://causal-learn.readthedocs.io/

To facilitate the comparison, we manually partitioned the MTS for Rhino.

| $d$ | Method | Type | $K = 2$ SHD | F1 | $K = 3$ SHD | F1 | $K = 4$ SHD | F1 |
|-----|--------|------|-----|-----|-----|-----|-----|-----|
| 10 | Rhino | Inst. | 36 | 36.4 | 48 | 37.8 | 64 | 39.1 |
| | | Lag | 38 | 36.9 | 60 | 33.6 | 85 | 34.7 |
| | CASTOR | Inst. | **0** | **100** | 1 | **97.3** | 1 | **99.3** |
| | | Lag | **0** | **100** | **0** | **100** | 2 | **98.4** |

Table 7: F1 and SHD Scores by Models and Settings: $d$ indicates number of nodes, $K$ refers to the number of regimes, and 'Type' refers to the causal links as either instantaneous or time-lagged.

In comparison to other baselines, Rhino surpasses VARLINGAM in terms of F1 score, underscoring its competitive performance in causal discovery.

Examining Table 7, it becomes evident that CASTOR consistently outperforms Rhino across various settings. Notably, CASTOR achieves this superior performance without relying on access to regime indexes, a noteworthy distinction. Moreover, Rhino necessitates a training phase that demands a larger amount of data to effectively learn the temporal causal graph.

### D.6 MODELS RUNNING TIME

We compute the running time of every model in different settings, that includes 10 nodes and 2,3 or 4 different regimes, Figure 6 summarizes the results.

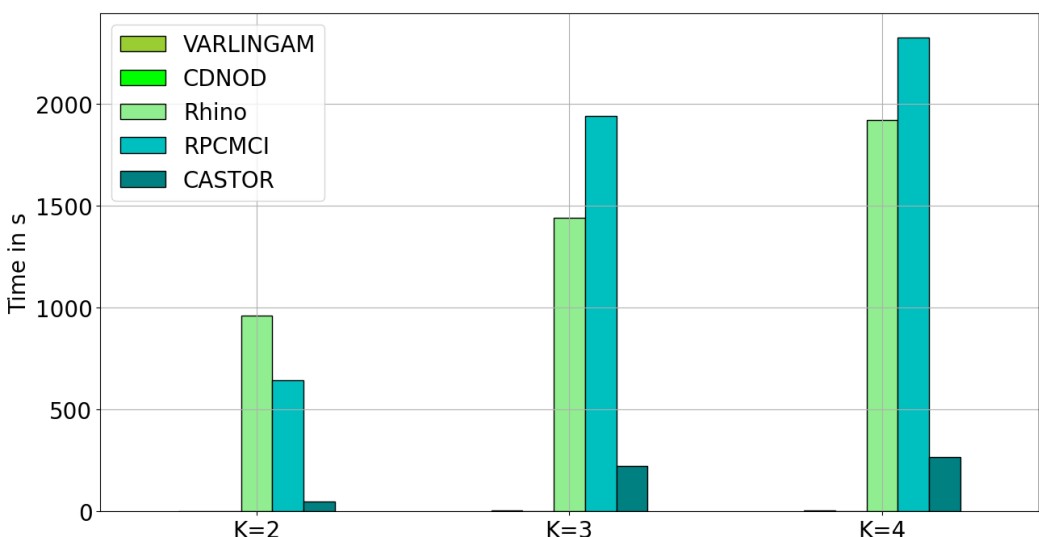

Figure 6: Running time per model, the y axis represents the running time in s and the x axis the number of regime

VARLINGAM and CD-NOD (employing Fisher's Z conditional independence test for faster runs; note that KCI CD-NOD takes over 2000 seconds for 2 regimes, with F1 scores in a similar range) exhibit remarkable speed compared to other methods. However, in terms of scores, both models encounter challenges in effectively learning causal graphs. Notably, CASTOR runs faster than Rhino, even though CASTOR learns both temporal causal graphs and regime indexes.

A fair comparison arises when comparing RPCMCI and CASTOR, as both models learn regime indexes and temporal causal graphs. It's essential to mention that RPCMCI necessitates specifying the number of regimes and the maximum number of transitions, producing only time-lagged relations. From Figure 6, it is apparent that CASTOR converges more rapidly than RPCMCI. This difference in convergence time becomes more pronounced as the settings become more complex.

### D.7 FURTHER EXPERIMENTS: NONLINEAR CASE

In this section, we present additional results using non-linear synthetic data with varying numbers of nodes, specifically $\{10, 20\}$, and diverse numbers of regimes $\{3, 5\}$. In this case, we compare our model against various baseline models, namely DYNOTEARS (Pamfil et al., 2020) and VAR-LINGAM (Hyvärinen et al., 2010). CASTOR showcases performance similar to DYNOTEARS in modeling non-linear time-lagged relations while simultaneously learning the regime indexes. Additionally, it effectively identifies instantaneous links. As depicted in figures 8 and 9, the performance of CASTOR diminishes with an increase in the number of regimes in non-linear scenarios. This outcome is understandable since non-linearity poses a more challenging causal discovery problem, and an increase in the number of regimes augments the number of parameters, consequently affecting our model's performance.

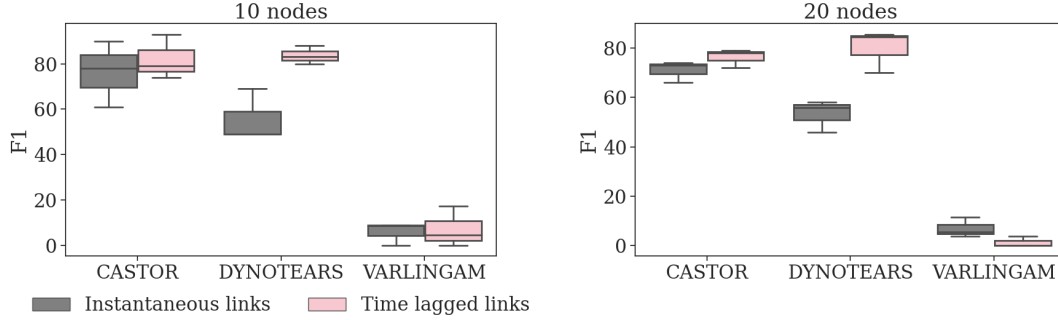

Figure 7: F1 by Models for K=3 Setting, Inst means instantaneous links and lag time-lagged ones.

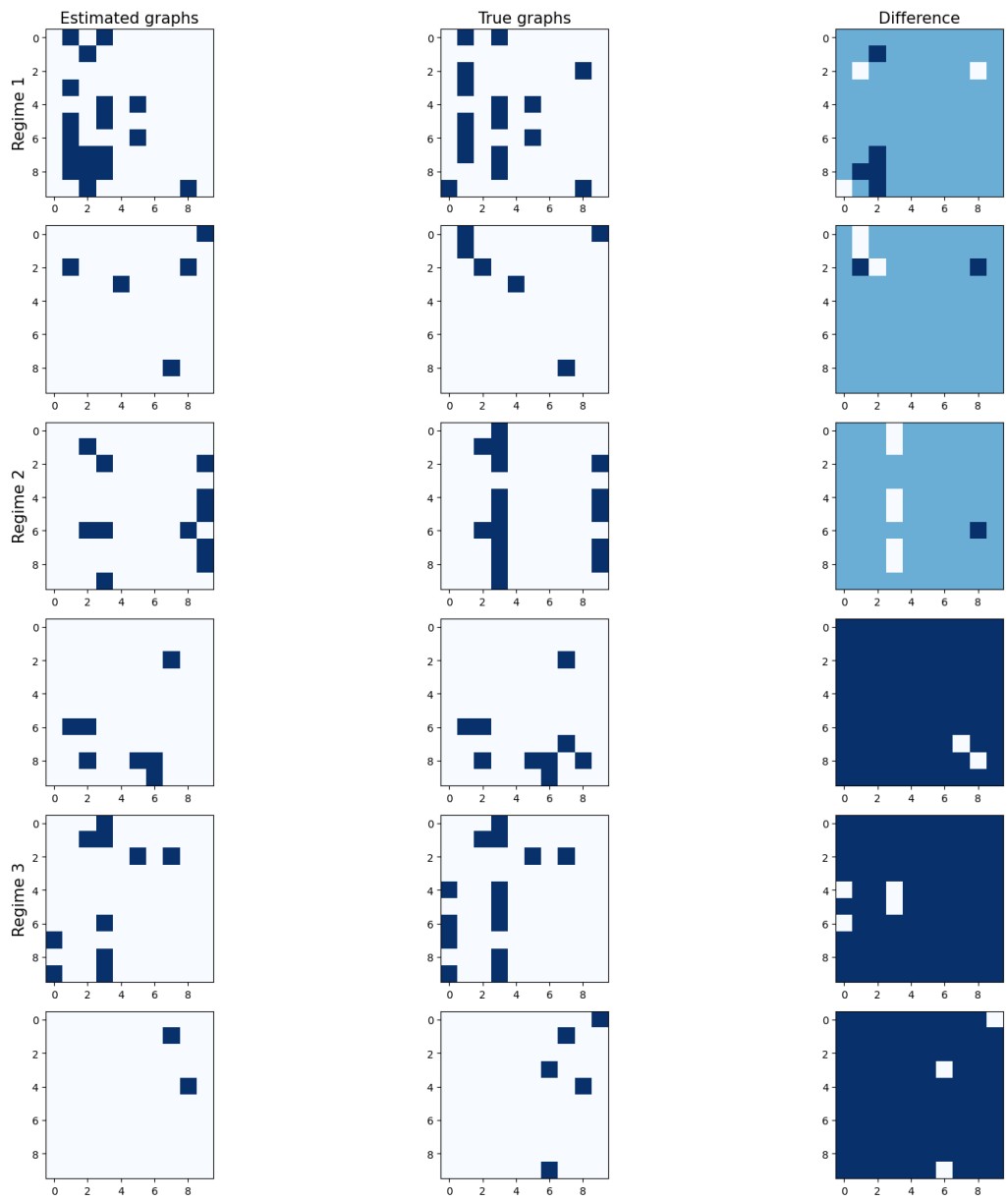

Figure 8: The estimated temporal causal graphs for 3 regimes **(non Linear case)**, with one matrix representing instantaneous links and another delineating time-lagged relations. The second column showcases the actual causal graphs, while the final column highlights the discrepancies between the estimated and true graphs.

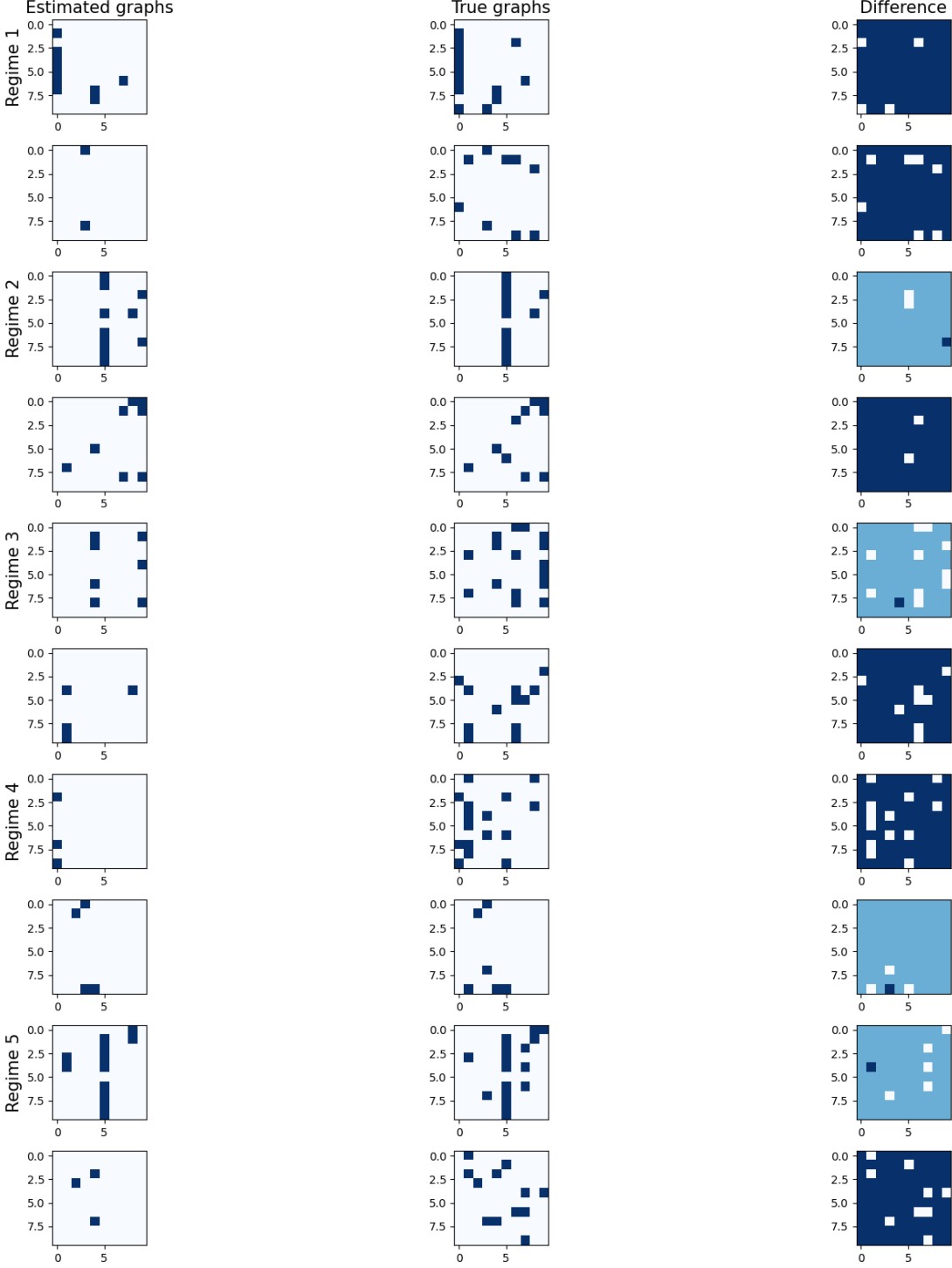

Figure 9: The estimated temporal causal graphs for 5 regimes (**non Linear case**), with one matrix representing instantaneous links and another delineating time-lagged relations. The second column showcases the actual causal graphs, while the final column highlights the discrepancies between the estimated and true graphs.

