# OpenReview forum: "Castor: Causal Temporal Regime Structure Learning"
_ICLR.cc/2024/Conference — Submitted to ICLR 2024_

### Official Review · Reviewer_Xsp2 · 2023-10-27

**Soundness:** 3 good
**Presentation:** 2 fair
**Contribution:** 1 poor
**Rating:** 5
**Confidence:** 4

**Summary:**

This paper tackles the following problem: given time series data that is mixed from multiple regimes, where the regime partition is unknown, the data in each single regime is generated by stationary additive noise temporal causal process, and the causal graphs in different regimes are different, the task is to recover the regime partition, and to learn the temporal causal graphs of each single regime.

This paper utilizes an EM method to maximize the data likelihood. Different from the original EM methods like (Zheng et al. 2018), this paper 1) introduces and learns an additional variable to model regime participation, and 2) allows nonlinear relationships by using neural networks like that in (Brouillard et al. 2020, Liu & Kuang 2023). The performance is validated on simulated and real-world datasets.

**Strengths:**

1. The technical derivations by using EM optimization to show the structure identifiability looks correct, though I didn't check the detailed proofs. The introduction of the regime participation variable is interesting.

2. The proposed method can handle instantaneous relations.

3. In experiments, the proposed method outperforms VARLiNGAM on each single regimes.

**Weaknesses:**

1. **The significance of the problem setting is unclear:**
   - It seems unnatural to assume that the time series data is from several mixed unknown regimes while within each regime the temporal causal relations are stationary. Could the authors give some motivative or real-world examples of this kind of datasets?
   - Usually it's more natural to think of nonstationarity in time series data as changing constantly but gradually, i.e., in the context of this paper, the regime partition is as fine as each single time stamp, while the causal relations (densities, edges weights) change smoothly across different time stamps.
   - Specifically, in this paper, by "time partition" (Definition 2), did the authors assume that each partition is a continent time series block? If yes, it should be stated clearly, instead of a vague "time partition"; if no, why, and any motivations? Also, did the authors assume/constrain any "similarities" between the causal graphs of different regimes?

2. **Even under this setting, the problem can already be well solved by existing methods:**
   - This paper mentions CD-NOD (Huang et al. 2020) but claims that "CD-NOD cannot infer individual causal graphs and necessitates prior knowledge about the number of regimes." But based upon my understanding on CD-NOD, this statement is generally incorrect, and the regime partition is actually readily obtainable from CD-NOD's abstract output.
   - By using the time stamp IDs as the surrogate variable, CD-NOD's phases I and II can output the abstract causal graph where the parents of each variable is identified as the union of all its parents in graphs from different regimes. Then, note that there is a phase III, kernel non-stationary visualization (Fig 12), which generally estimates the variability of the conditional distribution $p(Xi | union.parents(Xi))$ over the time index surrogate. The different regime partition is then readily available, i.e., the locations/times where changes happen are detected. Finally, with regimes recovered, any methods like PCMCI can be used in each stationary regime to obtain the graphs of each.
   - So what's the benefit of using the authors' proposed method than just using CD-NOD as above? Or please correct me if I am wrong regarding my understanding on CD-NOD or the problem.

3. **The parametric assumption is quite limited:**
   - The authors categorize the proposed CASTOR as score-based methods. Just for the convention, I would suggest to specify it as optimization-based methods (e.g., NOTEARS), to be distinguished from those real score-based methods, like GES.
   - Just like many existing optimization-based methods, this paper assume that the exogenous noise in the additive noise model follows a standard Gaussian distribution (namely, the uni/equal-variance assumption). This, however, is generally untestable and highly impractical in real-world scenarios. When the equal-variance assumption is violated, these optimization-based methods can perform relatively bad.
   - In this paper, is there any way to relax the uni/equal-variance assumption?

4. **The experimental comparisons are incomplete:**
   - The authors only provide experimental comparisons with stationary temporal causal discovery methods PWGC, VARLiNGAM, and DYNOTEARS. Moreover, the VARLiNGAM method is also misspecified for the non-Gaussian distributions.
   - More comprehensive comparisons are needed with at least some non-stationary temporal causal discovery methods, as well as the CD-NOD one discussed above.

5. **The presentation somewhat lacks clarity.**
   - What is the mixing coefficients $\pi_{t,u}(\alpha)$ in Eq.5? What is such $\alpha$? The readers are not expecting a choice of parameters (like the mentioned Same ́ et al. (2011)), but more from a definition perspective with physical meanings. E.g., in Eq.5, the mixing coefficients should be defined as something related to the regimes partition. Further parameter approximations are then introduced.
   - Why did the authors choose to spell out $Pa(<t)$ and $Pa(t)$ separately in expressions like Eq.2?
   - Question: does the proposed method output specific DAGs or equivalent classes? Since with instantaneous relations, usually the exact DAG is unidentifiable non-parametrically. Which is the critical factor that helps the exact identification in this method?

**Questions:**

See the Weaknesses part.

---

> ### Author Response · Authors · 2023-11-15
> **Response to Reviewer Xsp2**
>
> We extend our gratitude to Reviewer Xsp2 for the very insightful comments. We are pleased to note the appreciation for the introduction of regime learning and the acknowledgment of our approach in handling both instantaneous and time-lagged relations.
>
> **The significance of the problem setting is unclear:**
> - We are committed to providing further motivation for the problem at hand. In our introduction, we highlighted the relevance of our framework to EEG recordings in epilepsy, where distinct regimes (non-seizure, pre-ictal, seizure) form Multivariate Time Series (MTS) sequences. Additionally, Saeed et al.'s work, on causal structure discovery from mixtures of Directed Acyclic Graphs (DAGs) presented at ICML 2020, shows the importance of this setting in learning gene regulatory networks from diverse subtypes in neurological diseases. We believe that there are several other settings where our framework is particularly relevant and attractive, and we will revise the text to clarify these points.
> - Then, we clarify that each partition in our framework corresponds to a MTS block, representing a distinct regime. Visual aids in the form of figures have been added to the appendix to elucidate our setting.
>
> **Even under this setting, the problem can already be well solved by existing methods:**
>
> Our statement *"CD-NOD cannot infer individual causal graphs and necessitates prior knowledge about the number of regimes."*
> - We sincerely appreciate the insightful remark. Our statement aims to clarify that CD-NOD provides a summary graph for all regimes, which is a more constrained output compared to the comprehensive representation offered by our framework, CASTOR (Full temporal causal graph per regime). As the reviewer correctly noted, CD-NOD can infer change points but lacks the ability to express if regimes are repeated in the Multivariate Time Series due to the absence of temporal causal graphs for each regime. In contrast, CASTOR does not only learn and present the temporal causal graph for each regime but also leverages this information to achieve more accurate causal graphs by grouping together time series blocks from the same causal graph. This approach precisely determines the total number of distinct regimes.
> - We argue that learning a full temporal causal graph per regime is more powerful than acquiring a single summary graph. The full temporal causal graph does not only contain the information necessary to reconstruct the summary causal graph but also offers a more accurate evaluation metric. For instance, consider two simple settings with two variables: In setting 1, we have 1->2 in a time-lagged manner and 2->1 instantaneously, resulting in the summary causal graph 1<->2. Now, in setting 2, we have 2->1 in a time-lagged manner and 1->2 instantaneously, resulting in the summary causal graph 1<->2. However, evaluating based on the full temporal causal graph in each setting provides a nuanced and accurate assessment of the differences between them.
> - *what's the benefit of using the authors' proposed method than just using CD-NOD as above?* Thank you for your valuable feedback. We initially considered a fair and challenging comparison with methods learning full temporal causal graphs through manual regime splitting. We want to clarify that we had presented comparisons to RPCMCI which is a model for MTS with different regimes. However, in response to your suggestion, we conducted further experiments, comparing with CD-NOD and using phases 1 and 2 from the official Python package causal-learn, providing regime indexes to CD-NOD. The results of these extended comparisons are detailed below.
> |    | method | K=2     | K=3      | K=4      |
> |----|--------|---------|----------|----------|
> | 10 | CD-NOD  | 20.2    | 11.4     | 38.8     |
> |    | CASTOR | **100** | **100**  | **97.9** |
> | 20 | CD-NOD  | 25.2    | 23.7     | 12.7     |
> |    | CASTOR | **100** | **97.2** | **93.4** |
> | 40 | CD-NOD  | 0       | 11.3     | 5.57     |
> |    | CASTOR | **100** | **99.8** | **99.2** |
>
> **The parametric assumption is quite limited:**
> - We want to thank you for this remark, part of our future work will be to extend to more complicated settings with more complicated noises.
>
> **The experimental comparisons are incomplete:**
> - We have incorporated additional experimental results, conducting comparisons with CD-NOD and Rhino (CASTOR surpasses both models in various settings, achieving over 93% compared to 23% for CD-NOD and 39% for Rhino.). Please refer to the appendix for detailed outcomes ([Comparison with Rhino](https://openreview.net/forum?id=Qqu5mMgIBV&noteId=pdUoJO5frL)). Furthermore, we have included running times for all the models for comprehensive insights.
>
> **The presentation somewhat lacks clarity:**
> - We acknowledge the importance of clarity and have taken steps to address this concern. New figures have been added to the paper, available in the updated appendix, aiming to enhance the reader's understanding.

---

> > ### Author Response · Authors · 2023-11-15
> > **Response to the last question of Reviewer Xsp2**
> >
> > *Question: does the proposed method output specific DAGs or equivalent classes? Since with instantaneous relations, usually the exact DAG is unidentifiable non-parametrically. Which is the critical factor that helps the exact identification in this method?*
> >
> > We identify the DAGs. We used in our proof the theorem 1 in the work of Peters et al. (2013),"Identifiability of Gaussian structural equation models with equal error variances."

---

> ### Comment · Reviewer_Xsp2 · 2023-11-21
> **Thank the authors for your response.**
>
> Thank the authors for your response. And especially thank you for running additional experiments in short time.
>
> However, my two biggest concerns are still not very well addressed. Specifically,
>
>
> 1. **The significance of the problem setting is unclear:**
>
> > Motivative or real-world examples of such dataset, i.e., time series from several regimes where the relations inside each regime is stationary?
>
> The authors mention EEG recordings in epilepsy (Tang et al. 2021; Rahmani et al. 2023) and DAG mixtures in gene data (Saeed et al. 2020) to show the importance of this setting. However, for the former, could the authors please refer to the specific paragraphs in the two cited papers that explicitly model the EEG data as this setting? For the latter, if I understand correctly, they care about mixture of different DAGs in a same dataset without temporal partition, which is quite different from the setting of this paper.
>
> > Usually it's more natural to think of nonstationarity in time series data as changing constantly but gradually.
> >
> > By "time partition" (Definition 2), did the authors assume that each partition is a continent time series block? Also, did the authors assume/constrain any "similarities" between the causal graphs of different regimes?
>
> It seems that the authors didn't answer these questions. The authors did add some "visual aids in the form of figures" to the appendix, but here are some nit questions: in e.g., Fig 2, in each regime there are two adjacency matrices - one for instantaneous relations and one for time-lagged ones. How are time-lagged ones plotted in one matrix? Are the lags the same? Which colors of light and dark blues denote 1 or 0? Also, the y-axis 0.0, 7.5,... are misleading. Please make them integers.
>
>
> 2. **Even under this setting, the problem can already be well solved by existing methods (e.g., CD-NOD):**
>
> > By using the time stamp IDs as the surrogate variable, CD-NOD's phases I and II can output the abstract causal graph. Then, using kernel non-stationary visualization of phase III, the different regime partition is readily available.
>
> As mentioned earlier, it seems that what this paper does (identify different regimes, and then identify causal graph within each regime) can already be achieved by CD-NOD. So I am wondering what is the benefit of this paper.
>
> The authors mention "CD-NOD can infer change points but lacks the ability to express if regimes are repeated" -- I don't quite get it. What do authors mean by "repeated regimes"? I searched "repeat" in your paper but didn't find anything related. It seems that once each regime is identified, whether regimes are repeated will also be trivially available?
>
> The authors argue that "learning a full temporal causal graph per regime is more powerful than acquiring a single summary graph" -- that is unrelated. As mentioned above, the single summary graph is just a start point. CD-NOD can further identify the regimes partition and then temporal causal graphs within each regime can be estimated using many existing methods separately as well.
>
> Moreover, some of my other concerns still exists, e.g.,
>
> 3. **The parametric assumption is quite limited:** the authors have to assume the equal-variance of exogenous noise components, only with which the DAG identifiability under instantaneous relations is possible. However, this assumption is generally untestable. I don't put it as a major concern for this specific paper only because this assumption is widely used now by optimization based methods.
>
> In summary, my major concerns are not well addressed. So I keep my score of rejection.

---

> > ### Author Response · Authors · 2023-11-23
> > **Response to the reviewer (Part 1)**
> >
> > **Could the authors please refer to the specific paragraphs in the two cited papers that explicitly model the EEG data as this setting?**
> >
> > We appreciate the reviewer's feedback. In the cited papers, the authors employed graphs for epilepsy detection, yet the graph structure remained consistent for both non-seizure and seizure cases. These papers relied solely on node features to discern seizures or utilized graph-based correlations. Our proposal involves applying CASTOR to real-world settings, such as epilepsy, where the MTS comprises distinct regimes.
> >
> >
> > As an illustration, consider the Temple University Hospital Seizure (TUSZ) dataset, which is an open-source collection of recordings from epileptic patients. Each patient recording is a MTS characterized by multiple regimes, including non-seizure, pre-seizure, and seizure phases. Each regime represents a distinct MTS block lasting at least a few seconds, and a particular regime may occur more than once in the recording.
> > To elaborate, the patient typically begins in a stable state (regime 1 - non-seizure), where the brain exhibits normal electrical activity. Subsequently, an abnormal and excessive electrical discharge in a group of brain cells occurs, potentially leading to a seizure. This seizure regime (regime 2) persists for a few seconds or even minutes before the patient returns to their baseline, normal state (regime 1).
> >
> > The website of epilepsy dataset: https://isip.piconepress.com/projects/tuh_eeg/html/downloads.shtml#c_tusz
> >
> > **Did the authors assume that each partition is a continent time series block? Also, did the authors assume/constrain any "similarities" between the causal graphs of different regimes?**
> >
> > We thank the reviewer for the feedback, we mentioned in our reply that every regime is one MTS block. For this work we didn’t assume any constraints or similarity between the graphs.
> >
> > The non-smooth transitions in the graph between regimes are specifically driven by the unique dynamics of the epilepsy setting. The electrical discharge accompanying the onset of a seizure induces significant changes in the connectivity between EEG electrodes, leading to the disappearance of some edges and the emergence of new connections.
> >
> > In addition, the utilization of a temporal causal graph provides a more comprehensive explanation of seizure propagation. This approach incorporates instantaneous links, offering insights that can aid neuroscientists in localizing the seizure activity. Furthermore, it can contribute to distinguishing whether the seizure is focal, impacting specific brain regions, or generalized, affecting the entire brain.
> >
> > **How are time-lagged ones plotted in one matrix? Are the lags the same? Which colors of light and dark blues denote 1 or 0? **
> >
> > As for visual aids, the figures referenced can be found in our appendix, which was recently uploaded to the OpenReview website. Regarding the specific figure mentioned by the reviewer, we wish to clarify that these visuals were initially presented in the supplementary material accompanying our initial submission.
> >
> > In our submission, we include a visualization of the graph learned by CASTOR under various settings with a lag parameter (L) set to 1 (figure 4). Additionally, we provide another figure for L=2 (figure 5), where three matrices are presented for each regime (instantaneous, lag 1, lag 2). In these matrices, the dark blue color represents 1, while light blue represents 0
> >
> > **As mentioned earlier, it seems that what this paper does (identify different regimes, and then identify causal graph within each regime) can already be achieved by CD-NOD. So I am wondering what is the benefit of this paper.**
> >
> > Regarding the question of whether CD-NOD learns the temporal causal graph per regime, our understanding is that CD-NOD typically returns a causal summary graph for the entire Multi-Temporal Series (MTS). In contrast, CASTOR is designed to provide a distinct Temporal Causal Graph for each individual regime.
> >
> > We genuinely appreciate the reviewer's perspective, and while we hold great respect for their opinion, we would like to offer an alternative viewpoint. CASTOR not only excels at regime detection but also  captures the temporal causal relationships unique to each regime, setting it apart from CD-NOD. To illustrate, in the context of epilepsy recordings, CD-NOD may identify change points and generate a summary graph, but it might not explicitly reveal the occurrence of the non-seizure regime both before and after a seizure. CASTOR, by contrast, identifies the recurrence based on identical temporal causal graphs in the pre-seizure and post-seizure phases of the non-seizure regime.
> >
> > In our comparative analysis with CD-NOD, we applied a methodology where regime labels were assigned to CD-NOD. The results consistently highlight that CASTOR outperforms CD-NOD across various settings, as outlined in our previous response.

---

> > > ### Author Response · Authors · 2023-11-23
> > > **Response to the reviewer (Part 2/2)**
> > >
> > > **The authors mention "CD-NOD can infer change points but lacks the ability to express if regimes are repeated" -- I don't quite get it. What do authors mean by "repeated regimes"? I searched "repeat" in your paper but didn't find anything related. It seems that once each regime is identified, whether regimes are repeated will also be trivially available?**
> > >
> > > Thank you for your valuable feedback, particularly in the context of MTS composed of distinct regimes. Consider a scenario where a regime 'u' is repeated, as explained in the epilepsy setting above. CASTOR effectively groups all the diverse blocks of regime 'u,'  in the time partition $\mathcal{E}_u$ leveraging this repetition to enhance the accuracy of learning a temporal causal graph.
> > >
> > > Learning change points by CD-NOD and applying tools like Rhino or DYNOTEARS could incur higher computational costs in terms of running times.
> > >
> > > *Example:*
> > > To elaborate further, let's delve into the epilepsy setting: Imagine we have a recording from an epileptic patient where the sequence involves a non-seizure phase, followed by a seizure phase, and then a reappearance of the non-seizure phase. Employing CD-NOD, particularly with a KCI independence test, to detect regimes in such a scenario can be computationally expensive. Subsequently applying algorithms like Rhino, DYNOTEARS, or Lingam to learn temporal causal graphs would also be resource-intensive. This is primarily because the user would need to run the chosen algorithm at least three times (twice for non-seizure and once for seizure). In contrast, CASTOR concurrently learns the graphs and regime indexes, efficiently grouping the non-seizure indexes in one partition, leading to a more streamlined and resource-efficient learning process.
> > >
> > > **As mentioned above, the single summary graph is just a start point. CD-NOD can further identify the regimes partition and then temporal causal graphs within each regime can be estimated using many existing methods separately as well.**
> > >
> > > In reference to the comparison with baselines through manual regime splitting for different models, we want to clarify that all comparative assessments involving other baselines such as DYNOTEARS, Rhino, VARLINGAM were conducted by pre-splitting the regime manually for each model. Specifically, in scenarios where testing was performed on MTS with, for example, four distinct regimes, each constituting an individual MTS block occurring sequentially, we manually divided the MTS into four segments. Subsequently, we executed the models separately on these segmented time series.
> > >
> > > It is noteworthy that our comparative analysis consistently demonstrated CASTOR's superior performance over these models across various settings, including both linear and non-linear scenarios, even when the number of regimes increased to four or five.
> > >
> > > -----------------------------------------------------------
> > >
> > > We extend our gratitude to the reviewer for dedicating time to evaluate our work, and we highly appreciate their valuable feedback. While we respect the reviewer's perspective, we would like to express a respectful disagreement.
> > >
> > > In our rebuttal, we aimed to clarify that CASTOR, to the best of our knowledge, stands as a unique framework that not only detects regimes but also learns the Temporal Causal Graph per regime simultaneously. We respectfully assert that our approach offers distinct advantages, as highlighted in our responses:
> > >
> > > * CASTOR differs from CD-NOD in that it learns a temporal causal graph per regime, providing the exact number of regimes and their indexes. CD-NOD, on the other hand, focuses on a summary causal graph and change point detection without revealing the recurrence of regimes, thereby lacking the capability to determine the exact number of regimes.
> > > * Comparative analysis between CD-NOD and CASTOR on summary graphs consistently demonstrates CASTOR's superior performance.
> > > * We introduced a robust setting for comparison by manually splitting regimes for baselines and learning the temporal graph per regime. Our experiments convincingly show CASTOR's outperformance over these models. We argue that splitting manually regimes constitutes a more strong and robust evaluation than learning the regime indexes through CD-NOD.
> > >
> > > We express sincere appreciation to the reviewer for their valuable feedback and engaging in this constructive discussion.

---

### Official Review · Reviewer_VxPv · 2023-10-31

**Soundness:** 1 poor
**Presentation:** 3 good
**Contribution:** 2 fair
**Rating:** 3
**Confidence:** 4

**Summary:**

This paper proposes a framework named CASTOR, which is able to learn causal relationships from heterogeneous time series data composed of various regimes without prior knowledge of regimes.

**Strengths:**

- This method is designed to learn causal relationships from heterogeneous time series data.
- This work proposes a regime detection method.

**Weaknesses:**

- Theorem 1 states that each regime is identifiable, and thus the causal structure is also identifiable by giving the correct regime. However, the claim that the regime is identifiable is skeptical. For example, consider a simple normal mixture distribution with two components N(0,1) and N(0,2) (a degenerate case with two regimes), and this theorem seems to suggest that each sample can be identified whether from N(0,1) or N(0,2) which is clearly not true.
- Can you verify the identification of regimes with some extensive simulation study, especially for those that have distribution overlap in different regimes? For example, to show how is the accuracy in identifying the border (changed point) between two regimes.
- Moreover, is the number of regimes identifiable? How to choose the number of regimes.
- The task of identifying the regimes should also be related to the field of change point detection in time series which have not been comprehensively reviewed and compared with them.

**Questions:**

See the weaknesses above.

---

> ### Author Response · Authors · 2023-11-15
> **Response to Reviewer VxPv**
>
> We express our sincere gratitude to Reviewer VxPv for the dedicated effort and valuable comments on our manuscript. We are pleased that the reviewer acknowledges the quality of our paper's presentation and recognizes its strengths, particularly in regime learning and causal discovery from heterogeneous data.
>
> By addressing the concern raised by the reviewer regarding the correctness of our theorem, we appreciate the opportunity to provide clarification. The example presented by the reviewer, featuring two components with the same mean but different variance, neglects a critical aspect: all our noises originate from $\mathcal{N}(0,1)$. Consequently, variations in the mean of different components of the mixture occur when changing the graphs. This nuanced point ensures that if the means are the same, the underlying graphs are also identical, indicating the same regime. We are hopeful that this clarification will alleviate any doubts about the validity of our theorem.
>
> Concerning the inquiry about the choice the number of regimes, our paper explicitly addresses this problem in the learning algorithm. The detailed process is explained in the third section, involving the decomposition of the time series into equivalent windows. Subsequently, we iteratively update these windows by merging some, and canceling others. To enhance clarity, visual explanations, and figures detailing the learning process and CASTOR framework have been included in the appendix.

---

> > ### Comment · Reviewer_VxPv · 2023-11-22
> >
> > Thanks for the response. However, my concerns still exist.
> >
> > Regarding "variations in the mean of different components of the mixture occur when changing the graphs"
> >
> > First, there is no guarantee that the mean of different components must change when changing the graphs, and it is possible that each regime has the same mean.
> >
> > Second, even if the mean has changed, it is still unclear how to discern the sample from two normal distributions since there is always an overlap between two Gaussian distributions.
> >
> > Given the issues above, I keep my score of rejection.

---

### Official Review · Reviewer_4Xkb · 2023-11-01

**Soundness:** 2 fair
**Presentation:** 1 poor
**Contribution:** 2 fair
**Rating:** 3
**Confidence:** 3

**Summary:**

The paper presents CASTOR, a new framework for uncovering causal relationships in heterogeneous time series data. Unlike existing methods, CASTOR can identify different regimes within the data and learn the associated temporal causal graphs. This is achieved by leveraging the EM algorithm. The method is validated through synthetic experiments and real-world benchmarks.

**Strengths:**

1. The paper considers an important problem.

2. The framework is general to handle both linear and non-linear relationships.

**Weaknesses:**

1. The proposed method lacks novelty by comparing it with the NOTEARS method and the related extensions. See references below. For example, why not consider the nonparametric version of NOTEARS in the DYNOTEARS framework? Can you explain why combining EM is the most efficient way to solve the problem?

 - Zheng, Xun, et al. "Dags with no tears: Continuous optimization for structure learning." Advances in neural information processing systems 31 (2018).

 - Zheng, Xun, et al. "Learning sparse nonparametric dags." International Conference on Artificial Intelligence and Statistics. PMLR, 2020.

 - Pamfil, Roxana, et al. "Dynotears: Structure learning from time-series data." International Conference on Artificial Intelligence and Statistics. PMLR, 2020.

2. In addition, why consider or focus on the score-based method? One can not call the learned DAG a causal graph based on such type of method due to the absence of scale invariance.

3. I am very confused by the term "heterogeneous time series" in this paper. I cannot find any rigorous definition related to it. The "heterogeneous" in causal inference usually connects to heterogeneity among different individuals described by some variables say confounders. Anyway, without a clear statement, please do not claim such a contribution.

4. An analysis of computational complexity is needed or at least running time should be provided.

5. Many benchmark methods are missing for temporal causal graph learning. Please consider including at least methods mentioned in your related works.

6. Last but not least! The presentation needs substantial improvement. For example, one can find numerous flaws in Section 1. Please carefully check the citation formation and please do CAPITALIZE the first word in the sentence.

**Questions:**

Please consider addressing my comments in weaknesses.

---

> ### Author Response · Authors · 2023-11-15
> **Response to Reviewer 4Xkb**
>
> We extend our gratitude to the diligent efforts of the reviewer and value the insightful comments. We are grateful for Reviewer (4Xkb)'s recognition of the importance of our problem and the generality of our framework in addressing both linear and non-linear relations.
> The references cited by the author have been duly acknowledged in our paper, and they indeed represent special cases within our framework. We have added a discussion that permits to further clarify the novelty of the proposed framework.
>
> Notably, the reviewer references DYNOTEARS and its extension to the non-linear setting using the extended version of NOTEARS. We outline that our method encompasses these as special instances. In Equation 3 on page 3 of our paper, we present SEM of our settings. If $K=1$, corresponding to a time series with only one regime, it essentially becomes an extension of DYNOTEARS to non-linear settings. However, it is crucial to highlight that our framework operates in a setting where time series comprise multiple consecutive regimes, with each regime could be seen as MTS with a different causal graph (We employed the EM algorithm to iteratively alternate between regime learning and graph learning). CASTOR learns together the number of regimes, their respective indexes (indicating when each regime starts and ends), and the associated causal graph - none of the methods in the literature is able to address this joint learning problem, to the best of our knowledge.
>
> Thank you for the clarification. In our framework, heterogeneous time series refers to the fact that the time series data has multiple regimes. Based on our understanding, the term "heterogeneous data" or "heterogeneous time series" is used to denote data from different causal models in causal discovery literature (we provided some references). Your insight is valuable, and we have therefore refrained from using this terminology in the updated version of our manuscript to enhance clarity.
> - Saeed, Basil, et al. "Causal Structure Discovery from Distributions Arising from Mixtures of DAGs."  International Conference on Machine Learning. PMLR, 2020.
> - Huang, Biwei et al. "Causal Discovery from Heterogeneous/Nonstationary Data." The Journal of Machine Learning Research, 21
>
> To address concerns regarding runtime efficiency, we have incorporated comprehensive running time experiments in the appendix. Additionally, new models have been introduced, and all pertinent results are presented in the supplementary materials.
>
> **Setting: 10 nodes and different number of regimes (2,3,4)**
>
> VARLINGAM and CD-NOD (employing Fisher’s Z conditional independence test for faster runs; note that KCI CD-NOD takes over 2000 seconds for 2 regimes, with F1 scores in a similar range) exhibit remarkable speed compared to other methods (less than 4 seconds). However, in terms of scores, both models encounter challenges in effectively learning causal graphs. Notably, CASTOR (around 200 seconds for 4 regimes) runs faster than Rhino (around 2000 seconds for 4 regimes), even though CASTOR learns both temporal causal graphs and regime indexes.
>
> A fair comparison arises when comparing RPCMCI and CASTOR, as both models learn regime indexes and temporal causal graphs. It's essential to mention that RPCMCI necessitates specifying the number of regimes and the maximum number of transitions, producing only time-lagged relations. CASTOR (200 seconds for 4 regimes) converges more rapidly than RPCMCI (more than 2300 seconds for 4 regimes). This difference in convergence time becomes more pronounced as the settings become more complex.

---

### Official Review · Reviewer_eyWr · 2023-11-07

**Soundness:** 3 good
**Presentation:** 2 fair
**Contribution:** 2 fair
**Rating:** 5
**Confidence:** 3

**Summary:**

The paper presents a framework, called CASTOR, for causal discovery in heterogeneous time-series data across multiple regimes, each characterized by distinct causal graphs and functional relationships. The authors propose a combination of the EM algorithm with temporal structural equation models (SEMs) to simultaneously infer regime, causal graphs, and functional parameters. Theoretically, they demonstrate that by optimizing the proposed objective function, it is possible to recover the true regime and causal graphs, given certain assumptions hold. Empirically, the approach is validated through two synthetic experiments and application to two real-world datasets, showcasing the effectiveness of the proposed method.

**Strengths:**

This paper proposed a methodology that fuses the EM algorithm with SEMs to deduce regimes, causal graphs, and functional relationships from observational time-series data. There are precedents in the literature, such as Rhino, which employs an EM-like algorithm for inferring graph distributions and this paper's formulation bears a resemblance to both Rhino and DYNOTEARS. The introduction of multiple regimes in this context, however, does provide a new dimension to the existing frameworks. Although the degree of originality may not be profound, it is nonetheless a meaningful contribution to the domain. On the significance front, the paper targets a problem of practical relevance, potentially offering valuable insights and tools to the community engaged in causal discovery within time-series analysis. I have briefly checked the proof, which seems to be ok but there are some typos in the appendix.

**Weaknesses:**

The primary weakness of the paper is in its presentation, particularly regarding the explication of the key contribution: the introduction of multiple regimes. More textual emphasis and clarity on the parametrization of the time-series model to account for these multiple regimes would be advantageous. For instance, the ordering of time partitions denoted by $u_1$ and $u_2$ remains unclear – should the time in $u_1$ consistently precede that in $u_2$? Moreover, the role and interpretation of $\gamma_{t,u}$ are ambiguous. If $\gamma_{t,u}$ serves as an indicator for time $t$'s association with partition $u$, its continuous nature as presented in Equation 11 raises questions about how to understand the defined likelihood $\log p(X_{0:T})$ defined above Equation 8. A presentation from the perspective of variational inference might provide a more systematic and clearer framework, which could naturally elucidate derivations such as that of Equation 11.

Concerning empirical evaluations, the paper's baselines, namely VARLiNGaM and DYNOTEARS, do not represent the stronger baselines. Including stronger method, such as Rhino, could significantly enhance the demonstration of CASTOR's efficacy in regime inference. Matching or exceeding strong baselines would underscore CASTOR's advantages; alternatively, any performance gap would still furnish useful insights into the framework's relative standing and potential areas for enhancement.

**Questions:**

1. I wonder how should I interpret the generation mechanism for the time across the regime boundaries? For example, if we have two partitions: ${0,1,2,3}$ and ${4,5,6,7}$ with lag $1$ and $3$ respectively. Then, how should I interpret the data generation at $t=4$? What is the corresponding graph and lag I should use?

2. Do we assume the time in each partition always be smaller than the time in the later partitions?

3. How do you obtain eq.11? Is this the analytic solution from the E-step?

4. In appendix, Eq.22, what is $p^{(l)}$?

5. In appendix, is the $\pi$ introduced in Theorem 2 that same as the $\pi$ in the main text? If not, please use a different notation.

6. Missing equation number below Eq.22.

---

> ### Author Response · Authors · 2023-11-15
> **Response to reviewer eyWr**
>
> We sincerely appreciate the time and effort invested by the reviewer in evaluating our manuscript, and we are grateful for the insightful comments. We concur with the reviewer's observation that the introduction of multiple regimes adds a novel dimension to existing causal discovery frameworks.
>
> - Regarding the identified weakness in the paper's presentation, we acknowledge the importance of clarity and have taken steps to address this concern. New figures have been added to the paper, available in the updated appendix, aiming to enhance the reader's understanding. Additionally, we are refining the overall presentation and rectifying any typographical errors in the appendix.
>
> - In response to the specific query about $\gamma_{t,u}$, we want to clarify that we learn it, as detailed in Equation 11( E-step). Furthermore, in each iteration, the regime with maximum probability is assigned a value of 1, while others are set to 0.
>
> - In response to your observation concerning lag and regimes, our work operates under the assumption that the duration of the regimes is greater than the number of lags.
>
> - $p^{(\ell)}$ is the optimal distribution of the regime $\ell$.
>
> To address the comparison with the Rhino framework, we conducted rigorous evaluations. This involved training Rhino on 50 time series, each with a length of 200, for every regime. The regime indexes were provided to Rhino, and the entire process was carried out following the guidelines outlined in the authors' GitHub repository. We adhered to their README description to ensure accuracy and consistency in our approach.
> |     |        |      | $K=2$ |         | $K=3$ |          | $K=4$ |          |
> |-----|--------|------|-------|---------|-------|----------|-------|----------|
> | $d$ | method | type | SHD   | F1      | SHD   | F1       | SHD   | F1       |
> | 10  | Rhino  | I    | 36    | 36.4    | 48    | 37.8     | 64    | 39.1     |
> |     |        | L    | 38    | 36.9    | 60    | 33.6     | 85    | 34.7     |
> |     | Castor | I    | **0** | **100** | **1** | **97.3** | **1** | **99.3** |
> |     |        | L    | **0** | **100** | **0** | **100**  | **2** | **98.4** |

---

> > ### Comment · Reviewer_eyWr · 2023-11-22
> >
> > Thanks for the authors' response.
> >
> > I am a bit surprised that Rhino performs even worse the simple VARLiNGaM in this low-dimensional non-linear settings. Because from Figure 1, it seems that DYNOTEARS performs similarly to CASTOR, where in Rhino paper, Rhino seems to outperform DYNOTEARS with large margin. Perhaps, a more thorough comparison on all settings, e.g. nonlinear, Web, Netsim, is needed to confirm the real advantages of CASTOR.
> >
> > Therefore, I will keep my original evaluation.

---

> > > ### Author Response · Authors · 2023-11-23
> > > **Response to the reviewer**
> > >
> > > We sincerely appreciate the valuable feedback provided by the reviewer.
> > >
> > > In the current experimental setting, we have initiated a comparison between Rhino and CASTOR specifically in linear scenarios. It is worth noting that additional experiments are planned to comprehensively illustrate the advantages of CASTOR. In the preliminary comparison, we want to clarify that Rhino exhibited a superior performance over VARLingam based on F1 score.

---

### Author Response · Authors · 2023-11-15
**Response to all the reviewers**

We extend our sincere appreciation to all the reviewers for their invaluable feedback. We are pleased that our work has been recognized for presenting meaningful contributions to the field of causal structure learning (Reviewer [eyWr](https://openreview.net/forum?id=Qqu5mMgIBV&noteId=Gm7Uec8qIu) ) and addressing a crucial setting (Reviewer [4Xkb](https://openreview.net/forum?id=Qqu5mMgIBV&noteId=69xU3pkUZK)). The introduction of regime learning has been acknowledged as an interesting dimension of our method (Reviewiers [eyWr](https://openreview.net/forum?id=Qqu5mMgIBV&noteId=Gm7Uec8qIu) , [VxPv](https://openreview.net/forum?id=Qqu5mMgIBV&noteId=6F0DxTjPD1) , [Xsp2](https://openreview.net/forum?id=Qqu5mMgIBV&noteId=oO2mrrxClL) ), and our framework's capability to handle both instantaneous and time-lagged relations has been outlined as a new contribution.
The constructive comments of the reviewers have been carefully considered, and their main concerns have been diligently addressed in our revised draft. In particular, we have incorporated comparisons with stronger baselines such as Rhino (Reviewer [eyWr](https://openreview.net/forum?id=Qqu5mMgIBV&noteId=Gm7Uec8qIu)) and CDNOD for nonstationary time series (Reviewer [Xsp2](https://openreview.net/forum?id=Qqu5mMgIBV&noteId=oO2mrrxClL)), which further confirm the benefits of the proposed framework. Additionally, we have provided running time analyses for all the models (Reviewer [4Xkb](https://openreview.net/forum?id=Qqu5mMgIBV&noteId=69xU3pkUZK)). Typos in the manuscript have been corrected, and illustrative figures have been added in the appendix to enhance clarity regarding our settings.

---

### Author Response · Authors · 2023-11-23
**Manuscript revision**

We express our sincere appreciation to the reviewers for their thoughtful comments and valuable feedback. We have answered the main concerns of each reviewer directly as replies, but we have also made some changes to the manuscript we would like to highlight:

* We have addressed various typos, enhancing the overall clarity of the manuscript. Additionally, we introduced modifications to certain sections for improved comprehension. New figures have been incorporated to elucidate the settings, and the motivation behind our framework has been further refined.
* Recognising the importance of a comprehensive evaluation, we have conducted additional experiments by introducing Rhino as a baseline in one setting. It's worth noting that we are actively working on expanding these comparative settings.
* In response to reviewer feedback, we included a comparison with CD-NOD, demonstrating CASTOR's superior performance across different scenarios.
* To address concerns about efficiency, we performed a running time analysis, highlighting that CASTOR achieves accurate results with faster execution times compared to certain models.

We hope these changes can satisfy the requests of the reviewers and we remain open to any new requests they may have to improve our text.

---

### Meta-Review · Area_Chair_wjY2 · 2023-12-05

**Metareview:**

This paper considers time-series causal discovery across multiple regimes. The idea is to use an EM algorithm to simultaneously infer regime, causal graphs, and functional parameters of the SEM for each regime. Both theoretical guarantees (with assumptions) as well as empirical results are presented for justifications.

While reviewers agree that inferring regimes and performing regime-dependent causal discovery are important aspects for time-series causal discovery tasks, their main issues with this paper are:
1. Novelty and performance advantages over existing approaches, specifically Rhino and CD-NOD.
2. Concerns regarding the theoretical results, whether the assumptions are practical.

For the new results provided in revision, reviewers still have questions regarding (1) whether Rhino baseline's results are as expected, and (2) novelty of the approach over CD-NOD. Although the authors have promised more significant experiments in final version, I believe a rigorous scientific process cannot accept the camera ready version with significant and **non-peer-reviewed** changes over the submitted versions. Therefore, I believe another round of revise-and-resubmit will help strengthen the paper regarding comparisons over existing work.

**Justification For Why Not Higher Score:**

Questions still exist on the paper's presentation as well as its comparisons with Rhino & CD-NOD.

**Justification For Why Not Lower Score:**

N/A

---

### Decision · Program_Chairs · 2024-01-16

Reject